# Latent ion tracks were finally observed in diamond

H. Amekura [1] ✉, A. Chettah [2], K. Narumi [3], A. Chiba[3], Y. Hirano[3], K. Yamada[3], S. Yamamoto[3], A. A. Leino [4], F. Djurabekova [4], K. Nordlund [4], N. Ishikawa[5], N. Okubo[5] & Y. Saitoh[3]

Injecting high-energy heavy ions in the electronic stopping regime into solids can create cylindrical damage zones called latent ion tracks. Although these tracks form in many materials, none have ever been observed in diamond, even when irradiated with high-energy GeV uranium ions. Here we report the first observation of ion track formation in diamond irradiated with 2–9 MeV $C_{60}$ fullerene ions. Depending on the ion energy, the mean track length (diameter) changed from 17 (3.2) nm to 52 (7.1) nm. High resolution scanning transmission electron microscopy (HR-STEM) indicated the amorphization in the tracks, in which π-bonding signal from graphite was detected by the electron energy loss spectroscopy (EELS). Since the melting transition is not induced in diamond at atmospheric pressure, conventional inelastic thermal spike calculations cannot be applied. Two-temperature molecular dynamics simulations succeeded in the reproduction of both the track formation under MeV $C_{60}$ irradiations and the no-track formation under GeV monoatomic ion irradiations.

On nuclear fission of a uranium atom, fission fragments—i.e., a pair of high-energy heavy ions, one with a mass number of ~95 and the other with a mass number of ~140—are emitted with a total kinetic energy of ~170 MeV. When such high-energy heavy ions are injected into a solid, a huge quantity of energy of more than 10 keV nm$^{-1}$ is deposited along the ion trajectory. This high-energy deposition generates cylindrical regions of damage (typically a few nm in radius and around 10 μm in length for each fragment) called "latent ion tracks" (abbreviated hereafter to "ion tracks") along the ion trajectories in some materials; however, these ion tracks are not seen in all materials[1].

It is not only fission fragments that can form ion tracks: tracks can also be formed by high-energy heavy ions from large- and medium-sized particle accelerators[1]. High-energy heavy ions of between tens of MeV and several GeV, which dissipate their energy mostly via the electronic excitation in solids, are called swift heavy ions (SHIs). Over the past few decades, the formation of ion tracks by SHIs has been a central topic of research in ion–solid interactions[2–4] in various materials including metals[5], metallic compounds[6], oxide superconductors[7], semiconductors[8], inorganic insulators[9], and polymers[10].

The propensity for tracks to form depends strongly on the properties of the solids. Although there are some exceptions, in general, the propensity is higher for insulators[11] and lower for metals[11]; that of semiconductors is in-between[8,12,13]. Although tracks are formed in many insulators[11], one notable exception has been diamond. It is reported that neither fission tracks[14] nor ion tracks[1,15] have been observed to form in diamonds. The propensity for track formation is generally higher in *damaged* crystals than in corresponding undamaged crystals (which is known as the pre-damage effect[8]), because the damage reduces thermal conductivity and enhances electron–lattice coupling of solids[16].

However, even in a diamond that had been previously damaged by being irradiated with neutrons to a fluence of 2.5×10$^{20}$

[1]National Institute for Materials Science (NIMS), Tsukuba, Ibaraki 305-0003, Japan. [2]Department of Physics, LGMM laboratory, University of 20 Août 1955-Skikda, BP 26, route d'El Hadaiek-Skikda, Skikda 21000, Algeria. [3]Takasaki Institute for Advanced Quantum Science, National Institutes for Quantum Science and Technology (QST), Takasaki, Gumma 370-1292, Japan. [4]Department of Physics and Helsinki Institute of Physics, University of Helsinki, PO Box 43, FI-00014 Helsinki, Finland. [5]Japan Atomic Energy Agency (JAEA), Tokai, Ibaraki 319-1195, Japan. ✉e-mail: amekura.hiroshi@nims.go.jp

neutrons $cm^{-2}$, additional irradiation with 1.03 GeV Bi ions to a fluence of $2 \times 10^{10}$ ions $cm^{-2}$ induced neither track formation nor graphitization[17]. Zhang et al.-irradiated diamond powder with 1.4 GeV U ions to much higher fluences ranging from $5 \times 10^{12}$ to $8 \times 10^{13}$ ions $cm^{-2}$[18]. While a higher propensity was expected for diamond powder than the bulk crystals, the tracks were not formed in the powder, but the graphitization was detected.

The vanishingly low propensity for track formation in diamond, up to monoatomic GeV heavy ion irradiation, guarantees track-free diamond-anvil cells for applications involving SHI irradiation under high pressures[15,19]. SHI irradiation under high pressures is a powerful tool for studying the properties of minerals deep in the Earth's crust, and extremely high-energy SHIs are required to pass through a diamond cell of 2–3 mm in thickness. Although diamonds subjected to extremely high-energy U ion irradiation show ion penetration, no track formation has been reported. This low propensity for the track formation is also applied for the formation of the nitrogen-vacancy (NV) centers in diamond using monoatomic swift heavy ion irradiation intending the quantum applications[20].

Higher velocity SHIs emit higher energy δ-electrons toward the radial directions, which excite the cylindrical region along the ion trajectories. The radii of the cylindrically excited regions are determined by the range of the δ-electrons. Consequently, higher velocity SHIs form the excited regions of larger volumes because of longer δ-electron ranges, resulting in lower-density excitation. Lower velocity ions result in higher-density excitation, which is advantageous for track formation. We use the term "velocity effect" in this context[21,22].

Here, we report the first observation of track formation in diamonds with $C_{60}$ fullerene cluster ion irradiation of between 2 and 9 MeV, corresponding respectively to $S_e$ of between 29 and 52 keV $nm^{-1}$. Tracks were not observed under 1 MeV $C_{60}$ irradiation ($S_e = 21$ keV $nm^{-1}$). High-resolution scanning transmission electron microscopy (HR-STEM) observed the amorphization in the tracks and the partial formation of the crystalline graphite phase. The electron energy loss spectroscopy (STEM-EELS) mapping detected a π-bonding signal stronger in the tracks. Since diamonds do not show a melting transition at atmospheric pressure[23], the application of the conventional inelastic thermal spike (i-TS) calculations is not justified. Here, we applied the two-temperature molecular dynamics (TT-MD) simulations and succeeded in the reproduction of both the track formation under 2–9 MeV $C_{60}$ irradiation and the no track formation under GeV monoatomic ion irradiations. One of the advantages of the TT-MD simulations is no requirement of the input of the equilibrium melting temperature.

## Results

### The observation of ion tracks in diamond
Figure 1 shows bright field transmission electron microscopy (BF-TEM) images of irradiated diamond samples. As described in past literature[1,15], no one has succeeded in using monoatomic SHIs to form ion tracks in diamonds. Figure 1a illustrates this situation: A crystalline diamond was irradiated with monoatomic SHIs of 200 MeV $Xe^{14+}$; however, ion tracks were not observed, which was consistent with past literature[1,15].

In contrast, ion tracks were observed in diamond as black dots in Fig. 1b under 2 MeV $C_{60}^+$ ion irradiation. The areal density of tracks was $5.8 \times 10^{10}$ tracks $cm^{-2}$, which was comparable to the ion fluence of $5 \times 10^{10}$ $C_{60}$ $cm^{-2}$, indicating that one track was generated by one $C_{60}$ ion impact. Irradiation was at an incident angle of 7° from the surface normal[24]. The electronic stopping power $S_e$ of 29 keV $nm^{-1}$ for 2 MeV $C_{60}$ ions in diamond (which was determined as described in the "Methods" section), was identical to that of 200 MeV $Xe^{14+}$ ions ($S_e = 29$ keV $nm^{-1}$). Despite the identical $S_e$, tracks were observed in diamond irradiated with 2 MeV $C_{60}$ ions but not with 200 MeV Xe ions.

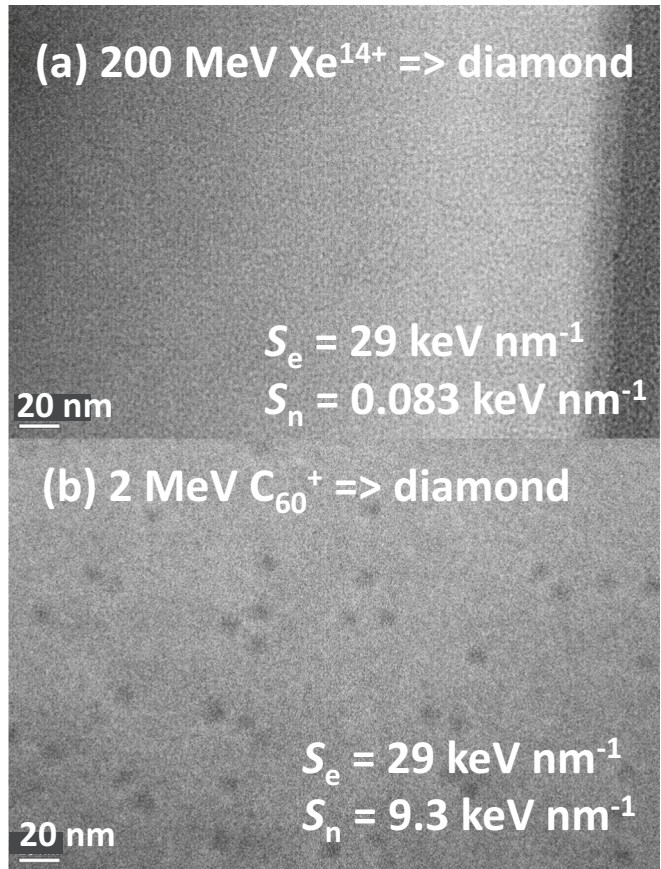

**Fig. 1 | Bright-field transmission electron microscopy (BF-TEM) images of diamond samples irradiated with (a) 200 MeV $Xe^{14+}$ ions and (b) 2 MeV $C_{60}^+$ ions.** Both images were recorded at the over-focus conditions. Although the electronic stopping power $S_e$ is identical with both irradiations, the ion tracks are formed only under $C_{60}^+$ ion irradiation. $S_n$ denotes the nuclear-stopping power.

By utilizing 2 MeV $C_{60}$ ions, we have succeeded in forming ion tracks in diamond for the first time.

### Shapes, dimensions, and the $S_e$ threshold of the tracks
Figure 2 shows BF-TEM images of diamond samples irradiated with 9 MeV $C_{60}^{2+}$ ions, with tilting angles of (a) 0° and (b) 30° (see the "Methods" section and Supplementary Methods 1 for the configuration). The tilting angle denotes the angle between the TEM observation direction and the sample surface normal. The samples were irradiated with the $C_{60}$ ions at an incident angle of 7° from the surface normal[24]. Figure 2a shows black dots that were later ascribed to ion tracks in diamonds and their overlapping surface craters/hillocks. We evaluated some of these dots under over- and under-focus conditions and found that the track images (dots) changed from black to white in accordance with the defocus condition (Supplementary Note 1). The contrast reversal indicates that the dots, i.e., the ion tracks, have a lower density than the medium[25,26]. As shown in Fig. 2a, the tracks observed from the surface normal direction are not perfectly circles but ellipses. This is probably due to the ion incident angle being 7° from the surface normal since the images of the track sidewalls overlapped with the track diameters (see Supplementary Note 2).

When the tilting angle is 30°, a long and narrow tail is observed to extend in the same direction from every ellipsoidal track-head (Fig. 2b). These long and narrow tails are ascribed to the track-bodies deeper inside the diamond. The track-bodies are visible with tilting, but not visible without tilting because the track-bodies are overlapped by the track-heads. The track-bodies are narrower than the

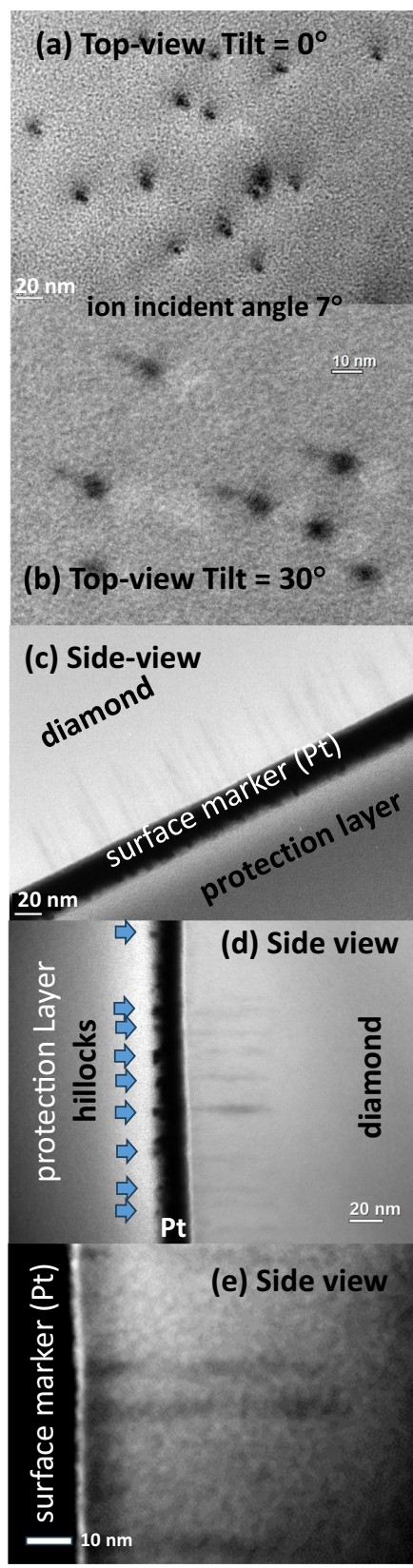

**Fig. 2 | BF-TEM images of diamond samples irradiated with 9 MeV $C_{60}^{2+}$ ions to fluences of $5 \times 10^{10}$ or $1 \times 10^{11}$ $C_{60}$ $cm^{-2}$.** Top-view images of irradiated diamond are shown with tilting angles of **a** 0° and **b** 30° from the surface normal. The $C_{60}$ ion irradiations were performed with an incident angle of 7° from the surface normal. Side-view images of ion tracks in **c** low-, **d** medium-, and **e** high-magnifications. Thick black layers in **c**–**e** are deposited Pt films as surface markers. In **d**, many protrusions are observed at the protection-layer side of the surface maker. While the observed protrusions are made of Pt, the origin of the protrusions can be the hillocks on the diamond surface, made by the ion tracks. **e** shows an expanded image of two tracks. All the images **a**–**e** were recorded at the over-focus condition.

irradiated with 40.2 MeV $C_{60}$ ions[29] and in $TiO_2$ with 1390 MeV Bi ions[30], where the track diameters decreased with the depth.

Figure 2c–e shows side-views of the tracks in low, medium, and high magnifications, respectively. Thick black layers are deposited Pt films for surface markers of ~20 nm in thickness, and the tracks are observed at one side of the Pt markers. To protect the Pt marker against the focused ion beam (FIB) milling for sample thinning, a thick carbon layer was deposited over it.

As shown in Fig. 2d, many protrusions were observed at the protection-layer side of the Pt marker. Judging from the image intensity, the protrusions are made of Pt. However, since the Pt layers were deposited *after* the irradiations with $C_{60}$ ions, the protrusions reflect the shapes of the irradiated surface, probably the shapes of the hillocks on the diamond surface. While the Pt thickness was thicker than the heights of the hillocks, the hillocks of quartz crystals have been successfully detected by this method[24]. A supporting observation of this method is the fact that each Pt protrusion is located on the extension line of each ion track.

It should be noted that a higher density of the tracks is observed in Fig. 2c than in Fig. 2d, probably because of the greater thickness of the sample of Fig. 2c. The higher density of the tracks results in higher density of the Pt protrusions. Because of the overlap of the protrusion images due to the higher density, the protrusions are not clearly distinguished from each other in Fig. 2c.

Figure 2e exhibits an expanded image of two ion tracks. Since this figure is not clear, one cannot judge whether cylindrical or conical tracks. Therefore, the observed tracks with narrower tails (Fig. 2b) could be ascribed to surface-craters[27], hillocks[28], or conical tracks[29,30]. As shown later, a track with three legs was observed by HR-STEM. To explain the strange track shapes, we assume the material emission from the track and deposition on the sample surface, which could be more consistent with the hillocks and craters than the conical tracks. From the indirect observation of the hillocks in Fig. 2d, the hillocks are most probable.

Comparing the mean track diameters determined from the top view with the 0° tilting and those determined from the side view for various $C_{60}$ energies (Fig. 3b) indicates that the mean diameters determined from the top views are always wider than the diameters determined from the side views. Error bars in Fig. 3b–d represent the mean ± the standard deviation (SD) of more than fifty tracks for each condition. The ion energy dependence of the mean track length is shown in Fig. 3c, indicating that the mean track length increases with the energy but is quite short, i.e., 52 nm even at the maximum energy of 9 MeV or less. The short track lengths are very different from those of SHIs.

As shown in Fig. 3d, the track formation threshold $S_{e,th}$ of 24 keV $nm^{-1}$ was determined by assuming the relationship[8]

$$R_{track}^2 = C(S_e - S_{e,th}) \qquad (1)$$

where $R_{track}$ and $C$ denote the mean track radius determined from the side views and a constant, respectively. Regarding the determination of the track diameters from the side-view images, see Supplementary Note 3. Since tracks were observed with 2 MeV $C_{60}$ ions

track-heads, indicating that the tracks consist of a two-step structure with two different diameters. Possible candidates are surface-craters[27] or hillocks[28] with wider diameters around the surface than the diameters of the tracks in depth, as depicted in Fig. 3a. Another possible candidate was conical ion tracks, which were observed in $Y_3Fe_5O_{12}$

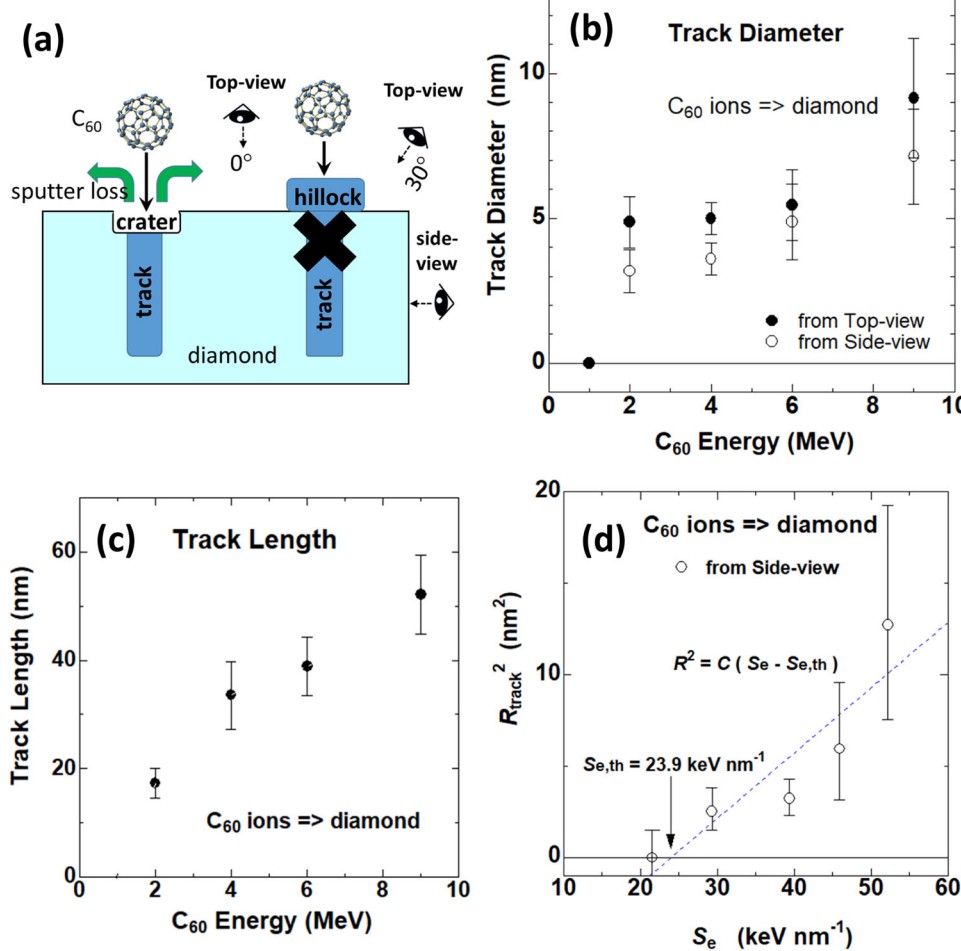

**Fig. 3 | Ion energy dependences of (b) the mean track diameter and of (c) the mean track length in diamond.** In **b**, the track diameters determined from the top-views (0°) and the side-views are indicated by closed and open circles, respectively. The difference between them indicates the two-step structures of the tracks, whose candidate structures are schematically shown in (**a**) as surface craters and hillocks.

In **d**, squared mean radii of the tracks determined from the side views are plotted against $S_e$, indicating the track-formation threshold of 24 keV nm⁻¹. In **b**–**d**, error bars represent the mean ± the standard deviation (SD), of more than fifty tracks for each condition.

($S_e$ = 29 keV nm⁻¹) and higher energies, but not with 1 MeV $C_{60}$ ions (21 keV nm⁻¹), the threshold value of 24 keV nm⁻¹ is justified. However, this threshold of 24 keV nm⁻¹ sounds too low because Khmelnitski et al. failed in the attempt to form ion tracks with 1.03 GeV Bi irradiation ($S_e$ = 41.0 keV nm⁻¹)[17]. As shown in the "Discussion" section, the TT-MS simulations indicate that this inconsistency can be ascribed to the velocity effect.

**High-resolution STEM observation**

Since ion tracks are formed more efficiently in graphite than in diamond[31], it is important to confirm that the tracks are formed in diamond and not in graphite. To obtain local information in the vicinity and inside of the tracks, HR-STEM was applied. From lattice fringes observed by HR-STEM, information on the crystallinity around the tracks is obtained, e.g., whether the tracks are formed in diamond or other phases, and whether the tracks are amorphous or not. For this purpose, we used a single crystalline diamond sample with a (100) face for the HR-STEM observations, because fine alignment among the track direction, the zone axis, and the electron beams was much easier using a single crystal. The sample was irradiated along the [100] zone axis with 9 MeV $C_{60}$ ions and observed along the same direction by STEM. Before the irradiation, the sample was thinned down to 36 nm thick, which is thinner than the mean track length of 52.1 nm with a standard deviation (SD) of 7.3 nm. Since the sample thickness

$x$ = 36 nm is less than the mean thickness minus 2 SD, the probability of an un-penetrating track P ($x$ < the mean thickness−2 SD) is 2.3%. Therefore, it is almost guaranteed that the deeper ends of all the tracks reach the other side of the sample. This quite thin thickness is important to judge whether amorphous regions form in the tracks or not, while the track formations in such a thin layer could be slightly different from those in bulk.

Figures 4a and b indicate STEM-BF images of ion tracks in diamond at (a) low and (b) medium magnification. While many tracks are observed in Fig. 4a, most of the tracks show white cores and black shells. Figure 4c is a fast Fourier transform (FFT) pattern of a high magnification image (not shown), and Fig. 4d was reconstructed from the signal inside the red circle in Fig. 4c by filtering the inverse FFT to stress the lattice fringes. While the lattice fringes of diagonal squares are observed outside the track, which is consistent with the diamond lattice, an amorphous region was observed in the track. Between the track core and the diamond matrix, black regions were observed. Since it is a STEM-BF image, the white core is ascribed to lower density or thinner thickness. Contrary the black track "shells" can be ascribed to higher-density regions or simply greater thickness. The high-angle annular dark field STEM (HAADF) observation was also carried out and confirmed these density/thickness changes as shown in Supplementary Note 4. While this is only speculation, the core became in lower density or lesser thickness due to material emission induced by the

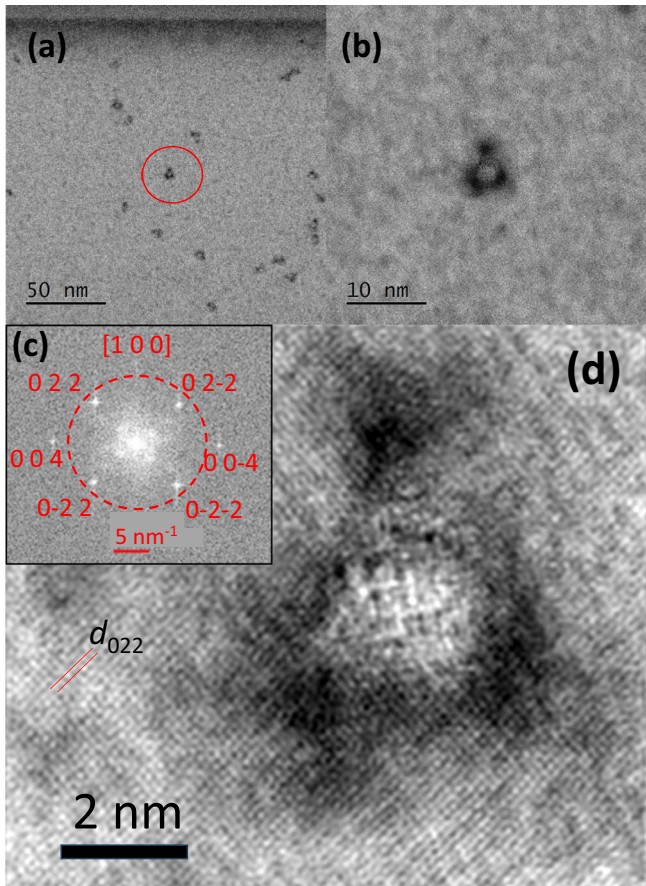

**Fig. 4 | Scanning transmission electron microscopy bright-field (STEM-BF) images of ion tracks formed under 9 MeV C$_{60}$ ion irradiation incident to the [100] zone axis of a single crystalline diamond.** The ion fluence was $5 \times 10^{10}$ C$_{60}$ cm$^{-2}$. **a** Low and **b** medium magnification images. **c** a fast-Fourier transform (FFT) image of a high magnification image (not shown). **d** was reconstructed from the signal inside the red circle in **c** by filtering the inverse FFT. The sample was thinned down to ~36 nm thick by FIB milling, which is thinner than the mean track length of $52.1 \pm 7.3$ nm (the mean $\pm$ SD), excluding the contribution of unirradiated bottom deeper than the track ends. One of the interplane distances is indicated in **d** as $d_{022}$. Since we used a TEM with the spherical aberration correction, the Scherzer defocus was not applied.

track formation. Parts of the emitted material were deposited at the surround of the track, and formed thicker regions, which are observed as the black peripheries of the track shown in Fig. 4d. However, further studies are necessary to confirm this speculation.

While Fig. 4c shows bright spots of 022- and 004-related diffractions from the diamond lattice, four diffuse spots are observed much closer to the center spot than the diamond D022 spots. These spots cannot be ascribed to the diamond but to the graphite G002 and G00-2. Since G002 and G00-2 appear as a pair of spots, the observation of the four spots indicates the existence of crystalline graphite grains of different orientations. HR-TEM observations show both the formation of amorphous regions at the track cores and of the crystalline phase of graphite.

## Chemical bonding inside the ion tracks

Since the possible existence of graphite in the track is suggested by the diffuse ring (spots) in the STEM-FFT pattern (Fig. 4c), it is important to know what carbon phase (graphite G or diamond D) is dominant inside the ion tracks. Since the mean track diameter (from the side view) is $7.1 \pm 1.3$ nm (mean $\pm$ SD) under 9 MeV irradiation, it is possible to use STEM-EELS to map the phase inside the tracks because the spot-size of

STEM-mode of JEM-2100F is 0.5 nm, i.e., much smaller than the mean track diameter.

Figure 5a shows a STEM image of ion tracks in diamonds observed under conditions optimized for the EELS measurements but not for the STEM observation. The purpose of this low-quality image (Fig. 5a) is to indicate that the line-scanning of STEM-EELS was performed along the green line shown. Figure 5b shows the EELS spectra around the C 1$s$ edge. Spectra from literature for graphite and diamond are shown at the bottom of Fig. 5b by blue and red circles, respectively[32]. The graphite phase shows a characteristic peak at 285 eV, which is ascribed to the transition from the occupied 1$s$ to an unoccupied π* level. Also, the graphite phase shows a plateau from 295 to 308 eV, while the diamond phase exhibits a valley at 303 eV and an isolated peak at 306 eV. Colored solid curve spectra shown above were detected at various distances $x$ from the track center from 0 to 30 nm along the line shown in Fig. 5a. While the intensity depends on the distance, always a faint peak is observed at 285 eV which is attributed to the π* peak.

The experimental EELS spectra $I(E)$ (thin colored curves) in Fig. 5b were fitted as the sum of the standard data of graphite $I_G(E)$ and diamond $I_D(E)$[32] by adjusting the graphite ratio $y$, i.e.

$$I(E) = yI_G(E) + (1 - y)I_D(E), \qquad (2)$$

where $0 < y < 1$. The optimized curves were shown by thin black lines and the optimized ratios $y$ were shown in Fig. 5b. The graphite ratio $y$ was the highest at the center of the track, which reached $y = 0.30$. The ratio $y$ decreased with the distance from the center of the track.

This indicates that non-negligible content of graphite or at least π-bondings exist in the track cores. This observation could be consistent with the observation of broad graphite spots in the FFT pattern shown in Fig. 4c. However, it should be noted that the EELS spectrum ($x = 0$ nm) in Fig. 5b was detected at the center of the track. Furthermore, the spot size of our EELS was 0.5 nm, i.e., much smaller than the track diameter of $7.1 \pm 1.3$ nm (the mean $\pm$ SD), and the tracks are connected to the other side of the samples. Therefore, the observed EELS spectrum ($x = 0$ nm) was detected from the severely irradiated region. Even though, the diamond phase of only 30% transforms to the graphite phase. The 70% are not transformed to graphite after a C$_{60}$ ion impact.

Figure 5c indicates the dependences of the π* peak intensity and of the graphite ratio $y$ along the distance from the track center: The π* peak has a maximum at the center of the track and decreases with the distance from the center up to 7 nm. Beyond the distance of 7 nm, the peak intensity becomes almost constant even up to 30 nm. The ratio $y$ shows almost similar dependence as the π* peak. While the constant intensity of the π* peak outside the track could be ascribed to damage introduced during the FIB thinning, further increment of the peak inside the track is definitely ascribed to the impact of a C$_{60}$ ion. Comparing the mean track radius of 3.6 nm for 9 MeV C$_{60}$ irradiation detected by TEM observations, the EELS line-scanning provided the damage zone radius of ~7 nm. However, this disagreement can be reasonable because the π* peak represents the distribution of the π bonds, while the tracks represent damage aggregates of nanometric sizes. The spatial distribution of the π bonds may expand wider than the radius of the tracks. Different track radii depending on the measurement methods, e.g., TEM, XRD, and Raman scattering, were reported by, e.g., Lang et al. [33].

As described, Zhang et al. irradiated diamond powder with 1.4 GeV U ions to much higher fluences ranging from $5 \times 10^{12}$ to $8 \times 10^{13}$ ions cm$^{-2}$, i.e., 100–1600 times higher than the present study of $5 \times 10^{10}$ ions cm$^{-2}$. Zhang et al. observed no track formation but the graphitization[18]. In this work, the track formation was induced with 2–9 MeV C$_{60}$ ion irradiations to the diamond. Partial transitions from sp$^3$- to sp$^2$-bondings, i.e., graphitization, were observed in the tracks in

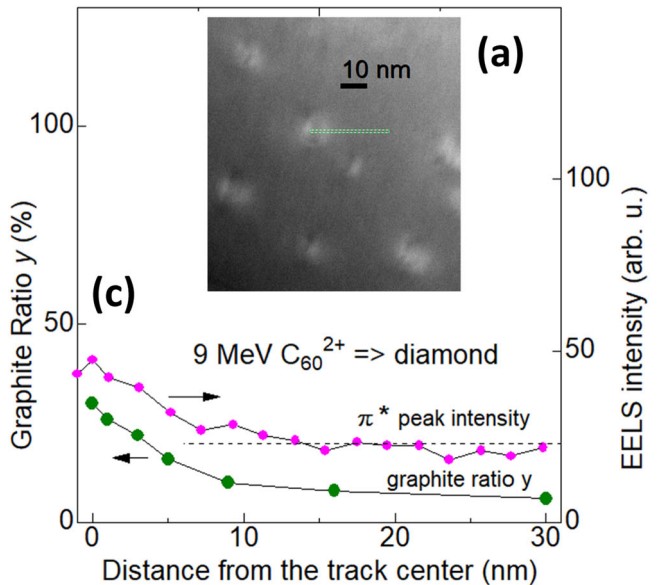

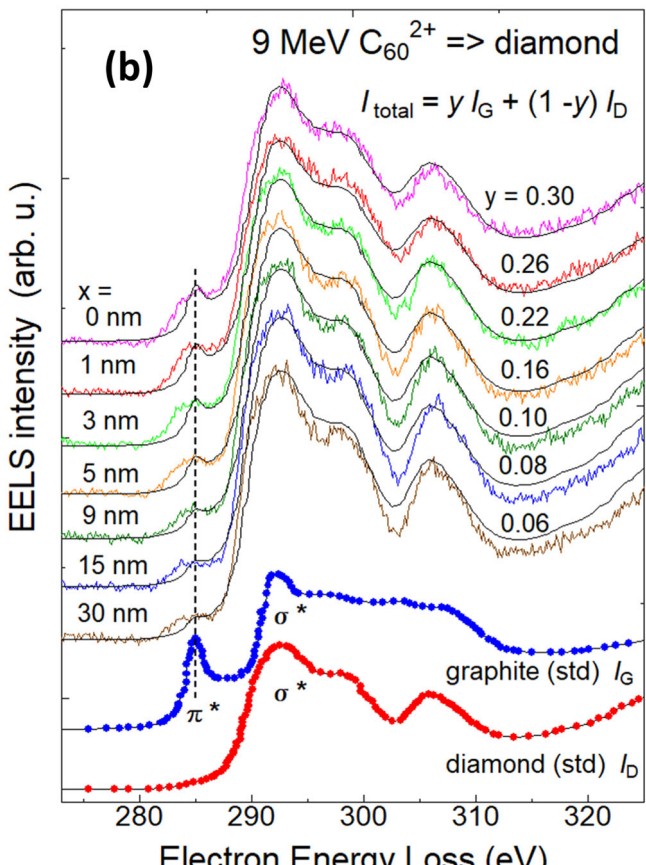

**Fig. 5 | Scanning transmission electron microscopy-electron energy loss spectroscopy (STEM-EELS) of an ion track in diamond.** The sample was irradiated with 9 MeV $C_{60}^{2+}$ ions at a fluence of $5.0 \times 10^{10}$ $C_{60}$ $cm^{-2}$. **a** A STEM image of ion tracks observed under conditions optimized for EELS measurements but not for STEM. The purpose of this low-quality image **a** is to indicate that the location of the line-scanning of STEM-EELS was performed along a green line. **b** EELS spectra at the C $1s$ edge. Blue and red spectra at the bottom are literature data of graphite and diamond, respectively[32]. Upper spectra were experimentally measured at various distances $x$ from the center of the track. Each spectrum was fitted by Eq. (2) as the sum of the standard spectra of graphite and diamond[32] with optimizing the graphite ratio $y$. The optimized curves are shown by thin black curves. The $\pi^*$ peak was indicated by a vertical broken line. **c** The intensity of the $\pi^*$ peak and the optimized graphite ratio $y$ are plotted along the distance from the center of the track. A horizontal broken line is a guide for the eye.

high pressure only. When heated at atmospheric pressure, the diamond does not melt but transforms to graphite at 2000–2200 K[23,34] and then sublimates itself around 4300 K[23,35]. Therefore, the application of the i-TS calculations is not justified to diamond. A more general approach is required: Here we have applied the two temperature molecular dynamics (TT-MD) simulations: This model describes the motions of C atoms interacting via the interatomic potentials under rapid energy deposition from the electronic system, without assuming any phase transitions such as the melting of diamond or graphitization (see the "Methods" section and Supplementary Note 5). This is in contrast with the i-TS model, where the phase transition temperature, mostly the melting temperature at the equilibrium, should be input. Furthermore, the TT-MD simulations could detect the track formation without phase transitions, i.e., without melting and vaporization.

Here a $C_{60}$ ion is modeled as an ion with high $S_e$ and quite low velocity. To compare the effect of the low velocity of $C_{60}$ ions and high velocity of GeV monoatomic ions, TT-MD simulations were carried out at two different velocities of 4.93 MeV $u^{-1}$ (GeV monoatomic ion) and 0.05 MeV $u^{-1}$ ($C_{60}$ ion) for three different $S_e$ (20, 40, and 60 keV $nm^{-1}$). Figure 6a and b show top- and side-views of simulation cells after the computation time of 100 ps from each ion impact. For the high-velocity cases (a), a track was not formed at 20 keV $nm^{-1}$. In the case of 40 keV $nm^{-1}$, the damage was localized close to the surface only as shown in the side view, indicating that it is no longer a continuous track. A continuous track was formed at 60 keV $nm^{-1}$. Contrary, a track was formed even at the lowest $S_e$ of 20 keV $nm^{-1}$ for the low velocity cases (b). With increasing $S_e$, the track diameter increased. The simulated tracks look almost spherical from the top, which is nearly consistent with low-magnification BF-TEM images (Figs. 1b and 2a) but not with high-magnification STEM images (Fig. 4).

The simulation and experimental results on the track diameters were plotted in Fig. 6c. Error bars in Fig. 6c represent the mean diameter ± SD of more than fifty tracks (experimental) and more than four regions of one track (simulations) for each condition. Red circles represent simulation results for the low velocity (0.05 MeV $u^{-1}$), which well reproduced the experimental track diameters (green circles) under $C_{60}$ ion irradiations between 2 and 9 MeV. An exception was at 20 keV $nm^{-1}$. While the simulated result shows the track formation, it has not been experimentally confirmed. However, apart from the fidelity of the simulation model, this difference could be ascribed to the fact that the destruction of $C_{60}$ molecules by the nuclear collisions[13,29] is not considered in the TT-MD simulations. Due to the nuclear collisions, the mean track length was shortened to 17 nm for 2 MeV irradiation as shown in Fig. 3c.

Khmelnitski et al. reported that tracks were not formed in diamond under 1.03 GeV Bi irradiation (4.93 MeV $u^{-1}$)[17]; this observation of the null tracks is indicated by a purple triangle in Fig. 6c. Since it was reported that tracks were not formed in diamond with U ion irradiation of any energy[15], the highest $S_e$ attainable by U ions, i.e., the maximum $S_e$ at the Bragg peak ($S_e$ = 49.3 keV $nm^{-1}$ at 1.1 GeV U, 4.62 MeV $u^{-1}$), is

addition to the amorphization. Furthermore, FFT patterns indicate the formation of crystalline graphite.

## Discussion

A broadly accepted model for ion track formation is the inelastic thermal spike (i-TS) model[22], in which the tracks are considered as a consequence of the melting/vaporization phase-transition (under the atmospheric pressure) of the target material induced by the high-density energy deposition from a penetrating ion. However, diamond behaves differently: While the i-TS model presumes the melting transition under atmospheric pressure, diamond melts under extremely

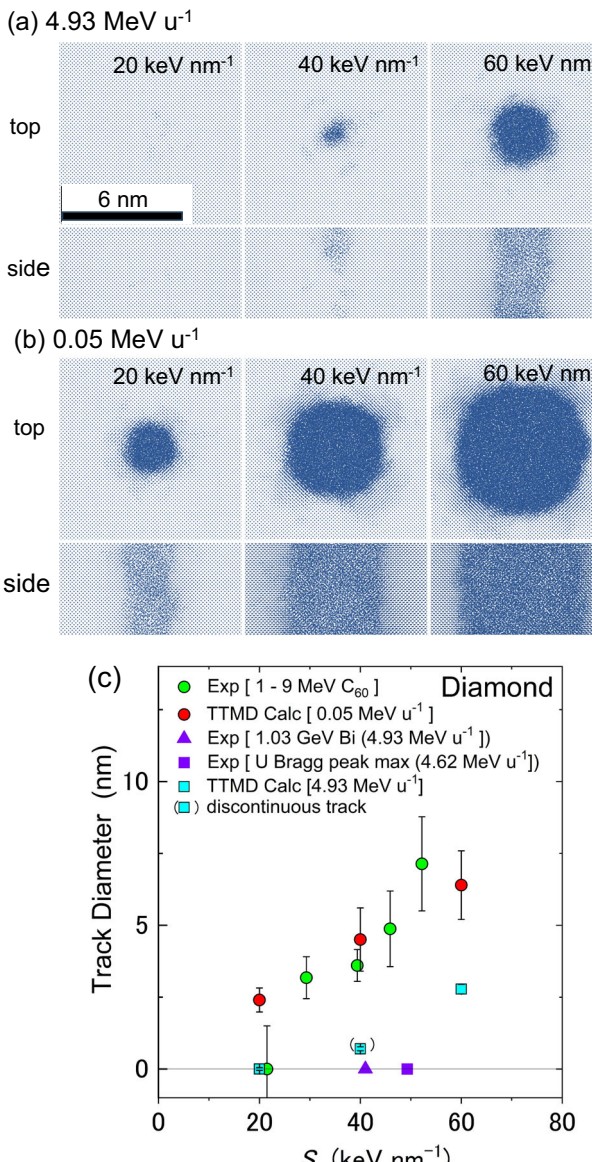

(a) 4.93 MeV u$^{-1}$

top — 20 keV nm$^{-1}$, 40 keV nm$^{-1}$, 60 keV nm$^{-1}$

6 nm

side

(b) 0.05 MeV u$^{-1}$

top — 20 keV nm$^{-1}$, 40 keV nm$^{-1}$, 60 keV nm$^{-1}$

side

(c) Diamond

Legend:
- Exp [ 1 - 9 MeV C$_{60}$ ]
- TTMD Calc [ 0.05 MeV u$^{-1}$ ]
- Exp [ 1.03 GeV Bi (4.93 MeV u$^{-1}$) ]
- Exp [ U Bragg peak max (4.62 MeV u$^{-1}$) ]
- TTMD Calc [4.93 MeV u$^{-1}$]
- discontinuous track

Y-axis: Track Diameter (nm)
X-axis: $S_e$ (keV nm$^{-1}$)

**Fig. 6 | Two-temperature molecular dynamics simulations of ion track formation in diamond.** Top- and side-views of simulation cells are shown for **a** high (4.93 MeV u$^{-1}$) and **b** low (0.05 MeV u$^{-1}$) velocity irradiations, at three different $S_e$ = 20, 40, and 60 keV nm$^{-1}$. The projectiles are injected at the center of the top surface, and the simulations were performed up to 100 ps after the ion impact. The visualizations display subsections of the simulation cell at the end of each simulation. Atoms are drawn as blue dots using an orthographic projection, and the top view is oriented in the [001] direction while the side view is in [010]. The simulated and experimental results on the track diameters were plotted in (**c**). Red circles represent simulation results for low velocity (0.05 MeV u$^{-1}$), which well reproduced the experimental track diameters (green circles) under C$_{60}$ ion irradiations between 2 and 9 MeV. The absence of track formation by 1.03 GeV Bi ions and by the highest $S_e$ of U ions are indicated by a triangle and a square in purple, respectively. The simulated diameters for the high velocity (4.93 MeV u$^{-1}$) are shown by cyan squares. Error bars represent the mean diameter ± SD of more than fifty tracks (experimental) and more than four regions of one track (simulations) for each condition. The visualizations were created using the OVITO PRO software[49].

plotted by a purple square (see Supplementary Note 6). As already described, the simulation results indicated almost no track formation at 40 keV nm$^{-1}$ for the high velocity, which is consistent with the experiments. While the track formation is predicted at 60 keV nm$^{-1}$ even for the high velocity, it is impossible to confirm, because the

highest $S_e$ attainable by a monoatomic ion is 49.3 keV nm$^{-1}$, which corresponds to the Bragg peak of U ion (purple square). Therefore, no track formation under high-velocity monoatomic ion irradiations is also reproduced by the TT-MD simulations.

Consequently, Fig. 6c has obviously clarified why ion tracks are *not* formed with swift monoatomic ions and why tracks are formed with slow MeV C$_{60}$ ions at the same $S_e$. The difference is due to the velocity effect. However, it should be noted that an approximation was provided to apply the TT-MD simulations to a C$_{60}$ ion impact. While each C atom consisting of a C$_{60}$ molecule is injected to a different position within the size of the C$_{60}$ molecule (the radius of 0.35 nm), sixty C atoms are approximated to be injected to the same position in the simulations. Since the radius in which 66% of the electronic energy is stored is 0.4 nm for each C ion in the present case, cooperative energy deposition is possible between C ions at different sites. However, this effect probably results in the enlargement of the excitation volumes, i.e., the reduction of the excitation density. The present simulations should overestimate the track radii. However, the experimental results were well-reproduced by the simulations. We cannot exclude other effects such as the synergy effect between $S_e$ and $S_n$ to enhance the track radii.

Therefore, the following scheme is proposed for track formation in diamonds. Tracks were formed under 2–9 MeV C$_{60}$ ion irradiation. The track formation threshold was estimated to be 24 keV nm$^{-1}$ for slow MeV C$_{60}$ ions and ~50 keV nm$^{-1}$ or higher for fast monoatomic ions. The different threshold is primarily ascribed to the velocity effect, which is supported by the TT-MD simulations. However, contributions from other effects cannot be excluded. Our success in forming ion tracks in diamonds is ascribed to the application of MeV C$_{60}$ ions, which are slow and have a lower track-formation threshold. HR-STEM observations and EELS mapping indicate that the track regions are amorphous and have more $sp^2$-bondings than outside the tracks. Broad diffraction of crystalline graphite was also detected by FFT of HR-STEM images.

## Methods
### Sample preparation
Poly-crystalline and single-crystalline diamond self-standing samples, both of which were grown by the chemical vapor deposition (CVD) technique, were purchased from Element Six Co. At first, the poly-crystalline samples were utilized for all the experiments, then the single-crystalline samples of 3 mm × 3 mm × 0.25 mm in dimensions were utilized for further HR-STEM and STEM-EELS measurements. The crystal face of 3 mm × 3 mm corresponded to the (100) of diamond, while the other faces of 3 mm × 0.25 mm corresponded to the (010) and (001).

Optical transmission spectroscopy in the ultra-violet and visible regions of the poly-crystalline samples was performed and verified the optical transparency up to the intrinsic bandgap energy of ~5.5 eV[36]. Also, the CVD growth guaranteed that the samples had little contamination from common impurities found in natural diamonds. Even in polycrystalline, the grain sizes were as large as several micrometers or more. While typical dimensions of the TEM samples were ~5 μm × ~10 μm, we seldom recognized the existence of grain boundaries during the TEM observations. See Supplementary Note 7 for more details.

TEM specimens were prepared in two different configurations (pre-thinned, thinned before ion irradiation; post-thinned, thinned after ion irradiation) to observe the top and side views of the ion tracks[13]. See Supplementary Methods 1 for details. The pre-thinned specimens were thinned down with 30 keV Ga focused ion beam (FIB) milling to a thickness of ~100 nm or thinner and were held on TEM grids. While the thickness of ~100 nm sounds thick, the low atomic number of diamonds assisted the electron transmission. A single-crystalline sample for STEM observations was thinned down to 36 nm in thickness. Then the pre-thinned specimens with the TEM grids were irradiated with C$_{60}$ ions at an incident angle of 0° or 7° from the surface

normal, to evaluate the top views of the tracks (the track diameter distributions). The single-crystalline sample for STEM observations was irradiated along the [001]. In the case of the post-thinned specimens, bulk CVD samples were irradiated with $C_{60}$ ions without thinning, and the cross-sectional samples were thinned down by FIB milling to observe the side views of the tracks (the track length distributions). To identify the surface position of the post-thinned samples, a thin layer of Pt was deposited onto the sample surface before FIB milling. The method was basically similar to what we had applied to crystalline silicon before[13].

Because of the much lower sputtering yield of diamond, more than ten times higher FIB fluence was required for thinning of the samples compared with Si. Since tracks are embedded in the surface layer of the post-thinned samples, the surface layer was protected by a thick layer of deposited carbon. Despite this precaution, track layers were sometimes lost due to the extremely high FIB fluences. The thinning process was quite time-consuming and required special care.

## $C_{60}$ ion irradiation

Irradiation of diamond samples with $C_{60}$ ions was conducted at the Takasaki Institute for Advanced Quantum Science, of the National Institutes for Quantum Science and Technology (QST), using a 3 MV tandem accelerator and a newly developed high-flux $C_{60}$ negative ion source[37]. Powder of $C_{60}$ with a purity of 99.5% was used. $C_{60}$ ions with a charge state of +1 ($C_{60}^+$) were utilized between 1 and 6 MeV irradiations, while those with a charge state of +2 ($C_{60}^{2+}$) were utilized for 9 MeV irradiation. Because of the magnetic mass separation, singly charged ions were free from contamination with fragments. However, doubly charged 9 MeV $C_{60}^{2+}$ ions could possibly be contaminated with fragments of singly charged 4.5 MeV $C_{30}^+$ ions, because of the same $m/q$ ratio. However, the amount of contaminating 4.5 MeV $C_{30}^+$ ions was negligible compared to the 9 MeV $C_{60}^{2+}$ ions. See Supplementary Note 8 for more details.

The samples were irradiated with low fluences of $5 \times 10^{10}$ or $1 \times 10^{11}$ $C_{60}$ cm$^{-2}$ to avoid creating overlapping tracks. The ion incident angle was set to 7° from the surface normal, except for a sample for HR-STEM observation; an incident angle of 0° was applied. For comparison, some samples were irradiated with 200 MeV $Xe^{14+}$ ions from the tandem accelerator at the Japan Atomic Energy Agency (JAEA), Tokai Research and Development Center.

The $S_e$ and $S_n$ of monoatomic ions were estimated from SRIM 2013 code[38]. Those of $C_{60}$ ions were estimated from the following relationship:

$$S_i(E, C_{60}) = \gamma_i N \cdot S_i(E/N, C_1) \tag{3}$$

where $i$ = e (electronic)[39] or $i$ = n (nuclear)[40], and $N = 60$ is presumed for $C_{60}$ ions. Although $\gamma_i \approx 1$ is frequently assumed in Eq. (3), the assumption of $\gamma_e \approx 1$ was recently bolstered by Kaneko's calculations[41]. Kaneko concluded that $\gamma_e$ is approximated as a constant of -0.8 between 2 and 10 MeV $C_{60}$ ions[41]. In this paper, $\gamma_e = 1$ is assumed for Eq. (3). For more details, see Supplementary Note 9.

## TEM observation

Three different transmission electron microscopes (TEM) were utilized for different purposes, while all operated at 200 kV. Bright-field TEM observation of the two configurations (pre- and post-thinned samples) was conducted using a JEOL JEM-2100 microscope. HR images were observed using a JEOL JEM-ARM200F with the spherical aberration correction since the $d$-value of diamond [220] is very small as 0.126 nm. Scanning TEM (STEM) and electron energy loss spectroscopy (EELS) mapping and the high-angle annular dark field STEM (HAADF) were conducted using a JEOL JEM-2100F microscope. For the EELS, the energy loss range between 230.0 and 497.8 eV was detected. To avoid the sample shift during the EELS measurements, the measurement

software compensated the shift with occasionally observing STEM images. To alternatively carry out both the EELS and STEM, a relatively long camera length of 20 cm was used for EELS measurements.

## Two-temperature molecular dynamics simulations

Simulations using the two-temperature molecular dynamics model (TTMD) are performed following the method by Ivanov and Zhigilei[42], which is implemented in PARCAS MD code[43-45] and described in ref. 46. It incorporates electronic effects into classical molecular dynamics with a friction term

$$F_i = -\nabla_i V(\{\vec{r}\}) + \xi v_i \tag{4}$$

where $F_i$ is the force acting on the $i$-th particle, $V$ is the original Tersoff potential (i.e., without the $\lambda_3$-term)[47], and $v_i$ is the velocity. The second term provides coupling with the electronic part of the two-temperature model[42],

$$C_e(T_e)\frac{\partial T_e}{\partial t} = \nabla(K_e(T_e) \cdot \nabla T_e) + G(T_e)(T_e - T_1), \tag{5}$$

$$T_e(t = 0) = A(r_\perp) \tag{6}$$

which is solved on a $51 \times 51 \times 1$ finite difference grid over the MD simulation domain (23 nm × 23 nm × 13 nm) in size. $T_e$ is the electronic temperature, $C(T_e)$ the electronic heat capacity, and $K_e(T_e)$ the electronic heat conductivity, holding the relation $K_e(T_e) = C_e(T_e)D_e$, where $D_e$ is the electronic heat diffusivity. $G(T_e)$ is the effective electron-phonon coupling and $A(r)$ describes the electronic temperature after the initial electronic collision cascade. The magnitude of the friction term $\xi$ is solved by requiring that the total energy of the system (heat equation + kinetic energy in MD simulation) is conserved at each timestep. More details are described in Supplementary Note 5.

The simulation cells are prepared with a relaxation run using the Berendsen thermo- and baro-stat[48] to 300 K, 0 GPa for 50 ps. The initial energy deposition is calculated so that the ion passes through the cell in the shortest direction (i.e., 13 nm), and the atoms near the four remaining outer boundaries are cooled to 300 K with a rapid Berendsen thermostat. To simulate the impact, the system is let to evolve for 100 ps. After this time, the resulting track radii are determined by measuring the size of the circular, darker regions from the top views in Fig. 6 by eye. At least four markers around each region are used to determine the radius and the error.

## Data availability
The datasets generated during the current study are available from the corresponding author on request. Source data are provided with this paper.

## Code availability
The code and software used in this work are PARCAS, OVITO, QUANTUM ESPRESSO, and SRIM which are openly available online from the corresponding developers and maintainers. OVITO PRO, which is not open source, was used to create the MD visualizations, but a similar analysis can be achieved with the open-source version.

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

## Acknowledgements

A part of the study was supported by the Inter-organizational Atomic Energy Research Program through an academic collaborative agreement among JAEA, QST, and the University of Tokyo. The authors are grateful to the crew of the accelerator facilities at QST-Takasaki and at JAEA-Tokai for their help. H.A. was supported by JSPS-KAKENHI Grant number 22K04990. TEM observation was performed using the facility of the NIMS TEM station. F.D. acknowledges Research Council of Finland SPATEC project #349690. We acknowledge CSC (Finland) for awarding this project access to the LUMI supercomputer, owned by the EuroHPC Joint Undertaking, hosted by CSC (Finland) and the LUMI consortium through development access program.

## Author contributions

K.Na., A.Chi., Y.H., K.Y., and Y.S. have developed the high-flux MeV $C_{60}$ ion beam. H.A. prepared samples. Y.S., A.Chi., Y.H., S.Y., and K.Na. conducted $C_{60}$ ion irradiation. N.I. and N.O. conducted 200 MeV Xe irradiation. H.A. conducted TEM observations. A.Che. carried out i-TS calculations. A.L., F.D., and K.No. carried out TT-MD simulations. All the authors joined in the discussion of the results and contributed to manuscript preparation.

## Competing interests

The authors declare no competing interests.
