## [Peer Review File · Nature Communications]

Latent ion tracks were finally observed in diamondREVIEWER COMMENTS

Reviewer #1 (Remarks to the Author):

The work, as far as I know, is well written, interesting and presents new results. Please here there are some recommendations to improve the readability of this really interesting work.

- 1) Please, include the Title "Introduction" where it corresponds.
- 2) In page 7 When describe Fig. 3, say that a typical period is 0.113 nm, could the authors draw marks showing the fringes in Fig. 3? They are difficult to find.
- 3) In Fig. 4d Why the authors assumes that the tracks shape are due to a densification of the tracks and not to geometrical effects due to a crater shape?
- 4) In Fig. 5a, The thermal spike is a very irreversible process, then the use of an "equilibrium phase diagram" could be, at least, controversial. I suggest a carefully use of equilibrium results for this kind of phenomena.

Reviewer #2 (Remarks to the Author):

This is a well written paper reporting important results which show unequivocally for the first time that so-called "(latent) ion tracks" can be formed in diamond under specific irradiation conditions. As the authors state, although such tracks have been previously observed in many insulators subjected to swift heavy ion (SHI) irradiation, diamond has hitherto been a notable exception. Although heavy monoatomic ions of up to GeV energies have failed to produce ion tracks, the paper demonstrates that these can be produced using C60 (fullerene) ions with energies from 2–9 MeV. The authors also provide a credible explanation of why the inelastic thermal spike model alone cannot account for their results and why the threshold (linear) measure of energy deposition differs for monoatomic and cluster ions. The paper thus merits publication and will be of interest to a broad set of readers with interests in swift heavy ions, radiation damage in solids, and the behaviour of diamond under extreme conditions.

There are a number of areas, however, where the manuscript could be made clearer:

The supplementary information provides images showing the contrast changes observed when using Fresnel (defocus) contrast to image the ion tracks but it is not clear whether the images presented in the paper itself have been recorded at focus or at over-focus – and this information needs to be indicated in the figure captions. I am puzzled that in the supplementary information, the tracks exhibit a darker contrast than the matrix at focus, as the defocus behaviour indicates that the inner potential of the tracks is lower than that of the matrix (consistent with the track core being lower density than the matrix as concluded from EELS measurements on p9). This should not yield a darker contrast at focus (except for amorphous material under specific conditions giving rise to structure-factor contrast). The question here, therefore, is whether there is residual dark contrast at focus and if so why?

The authors interpretation of Fig. 3 is not fully warranted and the figure caption should, in any case, contain more information on the imaging conditions for this phase-contrast image – in particular, the degree of defocus (e.g. from Scherzer defocus). From a single image as presented here, I don't believe it is possible to identify undulations, nor can it be concluded that "the ion tracks are not full of amorphous carbon but are probably damaged crystalline diamond". Note that I don't dispute this as an overall conclusion (specifically from the EELS analysis) – just that this cannot be concluded from Fig. 3.

Once again, in Fig. 4(a) insufficient information is contained in the figure caption. It is labelled "Spectrum Image" but the authors need to include the range of energies used to form the image (so that the reader can understand why the track appears as a white region in the image).

The explanation of the track formation via the high-pressure melting of the diamond seems credible but it is a little difficult to understand why subsequent epitaxial recrystallisation on cooling would leave a low density core surrounded by a higher density region. Perhaps molecular dynamics simulations could shed some light. (But this is a suggestion for further work and I am not suggesting that this should be included here).

The authors refer on a number of occasions to the "velocity effect" and contend that this provides the explanation of the different keV/nm thresholds for C60 and monoatomic ions. An explanation of the velocity effect can be found in their reference 21 and in references cited therein, but it would be useful to have a brief explanation of this (just a few lines) in the paper itself.

On a much more minor level:

Fig. 2(d) is probably superfluous – readers will understand the terms "crater" and "hillock" without this.

The statement on p5 that "the tracks observed from the surface normal direction are not perfectly circular but ellipsoids. This is probably due to the incident angle being 7° from the surface normal" cannot be correct given that the cosine of 7° is only 0.9925! And note that it should be "ellipse" rather than "ellipsoid".

Reviewer #3 (Remarks to the Author):

The authors made an interesting observation of ion tracks induced by C60 clusters in diamond. In general, ion tracks are easy to form in insulators but difficult to form in metals. However, diamond is one of the most unfavorable insulators to form tracks. The use of C60 clusters or high energy heavy ions (up to GeV) would significantly increase the chance to induce tracks in materials that were previously considered unfavorable for track formation, including metals and metallic compounds (Henry et al.,

NIMB 67 (1992) 390-395; Barbu et al., NIMB 145 (1998) 354). For example, the authors have recently reported track formation by C60 clusters in silicon (Amekura et al. Scientific Reports 11 (2021) 185). Although the irradiations by swift ions have been studied in diamond powders before (Zhang et al., NIMB 286 (2012) 262), ion tracks have never been observed in diamond. Thus, the observation of tracks in diamond by C60 clusters has important implications for understanding the track formation mechanism, which is still controversial. However, based on the problematic HR-TEM image shown in Fig. 3, the authors interpreted the inside of the C60 cluster induced tracks as the crystalline diamond. Consequently, the explanation of the high pressure of 0.1-10 GPa within the diamond during the track formation is also questionable.

The HR-TEM image in Fig. 3 is unable to clearly show the atomic structure in the inside of the ion track. The lattices in Fig. 3 appear diffuse, and the boundary between the track and the surrounding material is unclear. The possibility of the amorphous components within the track cannot be ruled out, as they are more difficult to determine than the crystalline components within the track, particularly when the quality of this image is poor. I understand that taking a high quality of HR-TEM image in diamond is not an easy task, as the track formation in this material is difficult, and an extended milling time by Ga ions is required for the sample preparation. For better HR-TEM images of ion tracks, the authors should satisfy some important requirements, including but not limited to, the very thin sample thickness, the minimum radiation damage during sample preparation, and the good alignment among ion tracks, zone axis and electron beams.

The authors failed to align ion tracks with the zone axis of the sample due to the use of the polycrystalline diamond and the 7° ion-incident angle from surface normal. As shown in previous high-quality HR-TEM studies (Vetter et al., 141 (1998) 747; Sachan et al., JMR volume 32 (2017) 928), the three directions, i.e., the electron beams, the incident ions, and a low index zone axis of the material should be strictly aligned to show sharp lattices and a clear track boundary with the matrix. As the investigated material is polycrystalline diamond in this study, however, it is practically impossible to align the ions along a known, low index zone axis prior to ion irradiations. During the TEM observation, it is also impossible to align ion tracks along the electron beams by tilting the zone axis of the sample, thus resulting in an elongated ion track, rather than a circular ion track, projected on the top surface.

The authors claimed that the inside of the ion track is damaged crystalline diamond, rather than graphite or amorphous carbon (Fig. 3). However, the sample was milled by extremely high fluence of Ga ions, which would induce significant radiation damage, more specifically the formation of amorphous carbon throughout the surfaces of the TEM sample. As a result, the Ga ions induced amorphous carbon on the surfaces would interfere the identification of amorphous carbon inside the track. As the amorphous components are more difficult to identify than the crystalline components directly from the HR-TEM image, the reviewer obtained the Fast Fourier Transform (FFT) images from two regions in the HR-TEM image of Fig. 3, that is, the center of the track (red frame) and a distance from the track (blue frame). As shown in SI Fig. 1 [*Editorial note: see below*], the diffuse ring in the center of the FFT image (bottom right) indicates more amorphous components within the track, while the absence of a diffuse ring in the center of the FFT image (upper right) indicates less amorphous components at a distance from the track. This distribution of amorphous component as obtained by FFT is against the authors' claim that the inside of the ion is crystalline.

SI Fig. 1 The Fast Fourier Transform (FFT) images from two different regions as marked in the HR-TEM image from the modified Fig. 3. The bottom right FFT image shows a diffuse ring in the center, indicating more amorphous components within the track (red frame). Such a diffuse ring is absent in the center in the upper right FFT image, indicating less amorphous components at a distance from the track (blue frame). This distribution of amorphous components is inconsistent with the authors' conclusion that the origin of the low-concentration graphite was not due to the track formation (Fig. 4b).

The formation of graphite-like carbon in the ion irradiated diamond powders was evidenced by the previous study of swift ion irradiation experiments, which clearly show a radiation-induced transformation to the graphite-like carbon by using electron diffraction, EELS, and Raman (Zhang et al., NIMB 286 (2012) 262). Raman is reliable to determine the graphite-like carbon, as the Raman signal is collected from many diamond grains, rather than a single grain in this study. The ring of graphite G(002) is also clearly identified from the electron diffraction from the diamond powders, for which the radiation damage is absent during the sample preparation. These results are contradicted with the interpretation of crystalline diamond in this study.

The sample appears too thick for HR-TEM image based on the diffuse lattices (Fig. 3). According to the Methods, the top view sample for HR-TEM was pre-thinned to ~ 100 nm before the ion irradiation. The Kikuchi lines (Fig. 2g) also suggest that a thick sample (at least 100 nm) was used for the electron diffraction pattern. A much thinner sample (20 nm or below) is usually required for a high-quality HR-TEM, although the authors may look for a region with a thinner thickness for better HR-TEM images. When the quality of the HR-TEM is not good, one should check if the sample is too thick. However, there is no other information for the sample thickness of the HR-TEM in the text.

Knowing that the orientations are unknown for the polycrystalline diamond, one may wonder why the authors avoided the ion channeling effect by choosing the 7° angle from surface normal. The authors have clarified that the large grain in Fig. S7 can be considered as a single crystal, although the diamond is

polycrystalline. For this HRTEM image, the zone axis can be determined as the [110] direction of diamond (SI Fig. 1), which is the same as the zone axis [110] in the diffraction pattern in Fig. 2g. However, this grain for the HRTEM image (Fig. 3) should be a different grain from the one in Fig. 2g due to a large angle ($\sim 70^\circ$) between their (1-11) planes in Fig. 2g and SI Fig. 1. Their choice of the 7° angle from surface normal can be validated only when all the grains are oriented along the same orientation (e.g., [110]).

The reviewer think that the crystalline components observed from the top-view HR-TEM are not solely from the inside of the cluster induced track. The crystalline components can be originated partially from the surrounding crystalline diamond, and/or at deeper depths of the sample. The track length is only 52.1 nm for 9 MeV C60 clusters and 17.3 nm for 2 MeV clusters, which are significantly shorter than the nominal thickness for the top view samples ~ 100 nm. In addition, the track width becomes narrower with increasing depths. All these allow the signals from the undamaged crystalline diamond at a deeper depth to project onto the crater-like tracks for the top-view HR-TEM images.

Other comments:

1. P1L2: High energy ions or fullerene ions create tracks not only in insulators, but also in metals or metallic compounds.
2. P2L7: In most solids, the total length of fission tracks is about 20 micrometers, and the length for each fission fragment is around 10 micrometers. The current statement is misleading. The readers may wonder if this length "more than micrometers" is the total length or the length along one direction.
3. P2L15: The authors should inform readers that ion tracks can be created in metals and metallic compounds, although tracks are normally easier to form in insulators.
4. P5L10: The best evidence to confirm the ion tracks is to compare the ion fluence with the track density, rather than using the reversal Fresnel contrast.
5. P6L2-6: The side view image (Fig. 2c) is not good enough to show the two-step structures. The best evidence is to show hillocks or caves from a high-quality side view image, such as in the paper by Shikawa et al., *Nanotechnology* 28 (2017) 445708.
6. P6L20: This is an orientation along [110] of cubic diamond, rather than a {1 1 0} plane. Fig. 2g also needs corrections for the orientation.
7. P7L10: The typical period is 0.206 nm for {1 1 1} planes, rather than 0.113 nm. The lattices with a distance of 0.206 nm can be measured from the HR-TEM image. The authors seem to confuse the concepts between planes and orientations. Refer to Fig. 2g, for which the zone axis is [110] direction rather than {1 1 0} planes.
8. P8L10: A peak around 295 eV, which is on the left of the 306 eV peak, can be ascribed to graphite. Amorphous graphite may form during the Ga ion sample preparation, or during the C60 irradiations.
9. P8L20: The EELS signal is related to, but not equal to, the density of the matter, as the electron energy loss peak also reflects variations in certain peaks of graphite or diamond. To show the core-shell structure, the authors should use the HAADF-STEM technique to show the Z contrast (or density) change.
10. P10L17: The calculation is inconsistent with the observation of amorphous graphite by Raman (Zhang et al., *NIMB* 286 (2012) 262). The formation of the amorphous graphite is not due to the D-G phase transition, but more likely due to the ion-irradiation induced formation of Frenkel pairs.
11. P11L17: Give references when the velocity effect was first introduced.

12. P14L1: The Ga ion thinning would induce additional radiation damage in the cluster ion damaged diamond. The Ga ion induced damage is difficult to separate from the damage induced by the cluster C60.

13. P14L10: The hardness of diamond may be related to the difficulties of sample thinning by Ga ions, but the dominant reason is the lower sputtering coefficient of diamond as compared to that of silicon.

Reviewer #4 (Remarks to the Author):

The manuscript describes ion track formation in CVD grown diamond films using C60 cluster irradiation. So far ion tracks have not been observed in diamond using single ion irradiation and I am not aware of any previous experiments with cluster beams. The investigation follows a similar paper from the authors on ion tracks in silicon produced by C60 clusters. Like in diamond, ion tracks had not been observed in Si following single ion irradiation.

The ion tracks were observed using TEM combined with EELS and calculations using a simple thermal spike model were performed to help explain the formation of ion tracks.

The main results from the experiments are:

- i. The formation of ion tracks using C60 clusters with energies between 2 and 9 MeV (no tracks were observed for 1 MeV clusters)
- ii. A track formation threshold of 23.9 keV/nm from linear fit to track data was established
- iii. The material in the tracks is crystalline (yet defective), no details on the nature of the defects was provided
- iv. EELS is consistent with a core shell structure of the tracks.

i. The main observation that ion tracks in diamond are formed using cluster beams is indeed interesting to the ion track community, although not very surprising. The manuscript provides some suggestions of how this can happen (in particular as irradiation with single ions of higher energy loss does not lead to track formation) but these are only speculations not backed up by experimental results. These suggestions include the velocity effect or synergies between electronic and nuclear stopping, both mechanisms have been discussed in the context of track formation previously.

ii. Often a threshold for the electronic energy loss for track formation is derived from ion track data and it has previously been argued that the threshold for some materials is higher than what can be obtained with single ions (thus track formation is not observed). In the current case I think the threshold value reported is not significant. Firstly clearly it is below the energy loss that has been achieved with single ions (where no tracks have been observed) and is thus only applicable for the particular cluster beam. Secondly it was derived from a simple linear fit to data that clearly looks not to be linear and thus will probably have a huge uncertainty (which has not been reported). The value is thus not very meaningful.

iii. The nature of the defects is not specified but it appears that the track is consistent with a trail of

defects. It may be difficult to provide more details on the nature of the defects but for an attempt on a mechanistic insight into track formation that would be important.

iv. The core shell structure is also not further explained. Could that be a surface effect resulting from the craters formed given the TEM samples are very thin? There is no indication of why this core shell structure is similar to what has been observed previously in (to my knowledge) amorphous materials

The thermal spike calculations use two simple heat diffusion equations with equilibrium values for the specific heat and thermal conductivities. While this model has indeed been used prolifically to describe aspects of track formation it is a gross oversimplification and has been heavily criticised (for example see S. Klaumunzer, *Mat Fys Medd Dan Vid Selsk* 52, 293 (2006)). Track formation is highly non equilibrium process that occurs on the nanoscale and the use of equilibrium quantities and phase diagrams not justified. Its success is mostly driven by its simplicity and the moderately good agreement with experimental track radii (which is not surprising in many cases as the electron-phonon coupling is often treated as a fitting parameter). While I agree that some interpretations of the model are useful to get insights in the track formation process, it cannot and should not be used to explain how tracks form. This is exactly what the authors do and my main criticism. For this reason, molecular dynamics simulations have superseded the i-TS model in more recent publications on mechanistic insights into ion track formation. The fundamental assumption of the authors is that either melting, sublimation or a transition from diamond to graphite need to occur (the model requires a local phase transition to define the track boundaries). There is no evidence of any phase transformation in the data, only defect formation. Thus, the entire interpretation that high pressure is generated and required for track formation is not substantiated.

In summary, while the general observation that ion tracks form in diamond upon cluster irradiation is certainly interesting to the ion track community, the manuscript does not provide any new insights into track formation nor can it explain the experimental observations other than using well discussed qualitative notion of the ion velocity and synergy between electronic and nuclear stopping.

Review Comments and One-to-one Response

Reviewer #1 (Remarks to the Author):

C1-0) The work, as far as I know, is well written, interesting and presents new results. Please here there are some recommendations to improve the readability of this really interesting work.

R1-0) Thank you very much.

C1-1) Please, include the Title "Introduction" where it corresponds.

R1-1) The manuscript formatting of Nature-Communications does not allow to include the heading "Introduction". After the abstract, the introduction should begin *without* the heading "Introduction".

C1-2) In page 7 When describe Fig. 3, say that a typical period is 0.113 nm, could the authors draw marks showing the fringes in Fig. 3? They are difficult to find.

R1-2) To response other Reviewers' comments, the HR-TEM image was exchanged with those taken from other sample with the [3 2 1] orientation. Figure R1(a) shows a HR-TEM image of irradiated diamond. According to FFT image (b), the fringes are ascribed

Fig. R1. (a) A HR-TEM image of diamond with the [3 2 1] orientation irradiated with 9 MeV C_{60} ions. The ion incidence was perpendicular to the surface normal. Fringe lines were observed from upper right to down left. According to FFT image (b), the fringes are normal to 1 -1 -1. Therefore, the distance of the fringes is ascribed $d_{1-1-1} = 0.2059$ nm. The observed periodicity was 0.22 ± 0.02 nm. (c) A corresponding simulated image by the multi-slice method using RiciPro software [R1].

to 1 -1 -1 spot. Therefore, the distance of the fringes corresponds to literature value $d_{1-1-1} = 0.2059$ nm, which is consistent with the observed periodicity was 0.22 ± 0.02 nm. Figure R1(c) shows a simulated image of HR-TEM by the multi-slice method using RiciPro software [R1], which is also consistent with the observation.

[R1] Seto, Y. & Ohtsuka, M. *Journal of Applied Crystallography* **55**, 397-410; (2022).

C1-3) In Fig. 4d: Why the authors assumes that the tracks shapes are due to a densification of the tracks and not to geometrical effects due to a crater shape?

R1-3) It is a good point. We have realized that the EELS intensity correlates with, but not exactly equals to the density of the materials. In fact, the EELS intensity variation was more than 20%, which is too high to be explained by the core/shell tracks. The variation could be due to the geometrical effects of crater shape. Since we cannot reach the firm conclusion of the core/shell track formation, we describe both the possibilities but don't specify one of them. As shown in Fig. R1(a), the central white part is the track (or core) and the irregular-shaped black part is the crater (or shell).

Following sentences were added in page 11 of the manuscript. "Possible candidate is (i) geometrical effects of the crater shapes or (ii) core/shell tracks [40,41]. Further study is necessary to clarify the origin."

C1-4) In Fig. 5a, the thermal spike (i-TS) is a very irreversible process, then the use of an "equilibrium phase diagram" could be, at least, controversial. I suggest a carefully use of equilibrium results for this kind of phenomena.

R1-4) Thank you. This is an important suggestion. We give up using the equilibrium phase diagram (old Fig. 5a) to discuss non-equilibrium processes such as the ion track formation.

Reviewer #2 (Remarks to the Author):

C2-0) This is a well written paper reporting important results which show unequivocally for the first time that so-called "(latent) ion tracks" can be formed in diamond under specific irradiation conditions. As the authors state, although such tracks have been previously observed in many insulators subjected to swift heavy ion (SHI) irradiation, diamond has hitherto been a notable exception. Although heavy monoatomic ions of up to GeV energies have failed to produce ion tracks, the paper demonstrates that these can be produced using C₆₀ (fullerene) ions with energies from 2–9 MeV. The authors also provide a credible explanation of why the inelastic thermal spike model alone cannot

account for their results and why the threshold (linear) measure of energy deposition differs for monoatomic and cluster ions. The paper thus merits publication and will be of interest to a broad set of readers with interests in swift heavy ions, radiation damage in solids, and the behavior of diamond under extreme conditions.

R2-0) Thank you very much.

There are a number of areas, however, where the manuscript could be made clearer:

C2-1a) The supplementary information provides images showing the contrast changes observed when using Fresnel (defocus) contrast to image the ion tracks but it is not clear whether the images presented in the paper itself have been recorded at focus or at over-focus – and this information needs to be indicated in the figure captions.

R2-1a) All the bright-field images shown in Fig. 1 and Fig. 2 were recorded at “the over-focus conditions”. This information was added in both the figure captions of Figs. 1 and 2.

C2-1b) I am puzzled that in the supplementary information, the tracks exhibit a darker contrast than the matrix at focus, as the defocus behavior indicates that the inner potential of the tracks is lower than that of the matrix (consistent with the track core being lower

density than the matrix as concluded from EELS measurements on p9). This should not yield a darker contrast at focus (except for amorphous material under specific conditions giving rise to structure-factor contrast). The question here, therefore, is whether there is residual dark contrast at focus and if so why?

A2-1) The images with the three different contrasts were observed in the sample irradiated with 6 MeV C₆₀ ions and are shown in Fig. R2. Care was paid to determine the in-focus conditions. At the in-focus condition, most of the tracks show much more diffused shapes than those at over- and under-focus conditions. Therefore, the tracks show almost no residual dark contrasts at the in-focus condition. While some minor exceptions exist, they exist close to the edge of the image. They are slightly out of the focal condition or may not be tracks. Maybe perfect focusing could be not easy.

This kind of focal condition dependence of ion tracks images have been reported before, e.g., Jensen et al. (NIMB 141('98)753). Unclear but weak images are observed at the in-focus condition.

C2-2b) From a single image as presented here, I don't believe it is possible to identify undulations, nor can it be concluded that "the ion tracks are not full of amorphous carbon but are probably damaged crystalline diamond". Note that I don't dispute this as an overall conclusion (specifically from the EELS analysis) – just that this cannot be concluded from Fig. 3.

R2-2b) We have carried out HR-TEM observation of *a new TEM specimen*, which was formed in more well-defined conditions:

- (i) The new TEM specimen has the surface which accidentally matches with the [3 2 1] orientation without any tilting.
- (ii) The specimen was irradiated at the incidence of the surface normal direction with 9 MeV C₆₀ ions. Therefore, the ion beam was introduced perpendicular to the specimen surface, i.e., parallel to [3 2 1].
- (iii) The HR-TEM observation was carried out perpendicular to the specimen's surface, i.e., parallel to the tracks. Mis-observation of the track sidewalls as the track interior is excluded.
- (iv) The sample was thinned down to 36 nm in thickness, which is thinner than the mean track length of 52.1 ± 7.3 nm. The thickness of 36 nm is thinner than the mean (52.1 nm) – 2 standard deviation (SD) (2×7.3 nm) = 37.5 nm. Most (more than 2SD >

97.7%) of tracks reach the other side of the specimen. Unirradiated deeper parts of the tracks are rare.

(v) We have observed more than ten tracks. All the tracks show the lattice fringes inside.

The probability of the un-reaching tracks is $(2.3\%)^{10} = 0\%$.

Therefore, the lattice fringes in tracks are not spurious but real.

The results are shown in Fig. R3. Please refer the caption, too.

C2-2a) The authors interpretation of Fig. 3 is not fully warranted and the figure caption

should, in any case, contain more information on the imaging conditions for this phase-contrast image – in particular, the degree of defocus (e.g. from Scherzer defocus).

R2-2a) More information was added in the caption of Fig. R3.

Fig. R3. High-resolution transmission electron microscopy (HR-TEM) images of ion tracks in diamond, formed under 9 MeV C₆₀ ion irradiation to the surface normal. The fluence was 5×10^{10} C₆₀/cm². The sample surface was accidentally aligned to the [3 2 1] orientation of diamond without any tilting. The sample was thinned down to ~36 nm thick by FIB milling, which is thinner than the track lengths of 52.1 ± 7.3 nm, excluding the contribution of unirradiated layers deeper than the track ends. While the Scherzer defocus $df_{sch} = -(4/3 C_s \lambda)^{1/2}$ was -57.9 nm for JEM-2100F (the spherical aberration coefficient C_s of 1.0 mm and the electron wavelength $\lambda = 2.5079$ pm), the defocus of -84 nm was applied for better images. **(a)**: HR-TEM image of four tracks. The lattice fringes are recorded all over the image (a), while it is not clearly shown in the image. **(b, c)**: Magnified images of two different ion tracks. Whole the regions of the inside the tracks show the lattice fringes. The fast Fourier transformed (FFT) images of **(d)**' the region on a track and of **(e)**' the region without tracks are shown in **(d)** and **(e)**, respectively. Line profiles of the FFT images along the diagonal lines shown in (d, e) are plotted in **(f)** with a vertical shift. Except the sharp central direct spot, both the profiles were almost identical with each other. The calculated lattice fringes by the multi-slice method using ReciPro code are shown in **(g)**. The experimental fringe period $d_{1-1-1} = 0.22 \pm 0.02$ nm as shown in (b), which matches to the literature value $d_{1-1-1} = 0.2059$ nm.

The Scherzer defocus $df_{sch} = -(4/3 C_s \lambda)^{1/2} = -57.8$ nm because spherical aberration coefficient $C_s = 1.0$ mm for JEM-2100F working at 200 kV ($\lambda = 2.5079$ pm). However, the images were observed with the defocus of -89 nm, which seems to provide the better images. As shown in Fig. R4, the fringe patterns were calculated at various values of the defocus and the sample thickness by the multi-slice method using ReciPro code [R1].

However, all the simulated fringes were similar with each other. The images do not strongly depend on the defocus and the sample thickness.

At $df = -87.8$ nm, thinner lines are observed at the sample thickness of 46 nm than those of 26 nm and 36 nm. These observations also support the sample thickness of 36 nm determined by the secondary electron microscopy (SEM) image.

[R1] Seto, Y. & Ohtsuka, M. *Journal of Applied Crystallography* **55**, 397-410; (2022).

C2-3) Once again, in Fig. 4a insufficient information is contained in the figure caption. It is labelled “Spectrum Image” but the authors need to include the range of energies used to form the image (so that the reader can understand why the track appears as a white region in the image).

A2-3) We apologize that the label “Spectrum Image” was misleading, while we intended that the spectrum image was observed from this HAADF image area. We deleted the label “Spectrum Image” to avoid confusion.

Figure 4a is a conventional HAADF image using a HAADF detector, not using an EELS detector. Since we didn’t use the EELS detector but a HAADF detector, we cannot

specify the range of the electron energies used for imaging. Furthermore, in order to detect the HAADF images and EELS spectra, a camera length of 20 cm was used, which is too long for pure HAADF images. Therefore, Fig. 4a is not a pure HAADF image, but was used to avoid the position shifts during the EELS mapping.

In the caption of Fig. 4, following sentence was added: “A HAADF image of ion tracks with a relatively long camera length of 20 cm.” and “While the EELS spectra between 230.0 eV and 497.8 eV were detected, only the limited regions are shown.”

C2-4a,b) The explanation of the track formation via the high-pressure melting of the diamond seems credible but it is a little difficult to understand why subsequent epitaxial recrystallisation on cooling would leave a low density core surrounded by a higher density region. Perhaps molecular dynamics simulations could shed some light. (But this is a suggestion for further work and I am not suggesting that this should be included here).

R2-4a) Reviewer #1 suggested the geometrical effects of crater shapes to explain the position sensitive EELS signal instead of the core/shell tracks. In fact, HR-TEM observed corresponding structures as shown in Fig. 3b and 3c. The central white parts could be the tracks (or cores) and the irregular-shaped black parts could be the crater (or shell).

We have realized that the EELS intensity correlates with, but not exactly equals to the density of the materials. In fact, the EELS intensity variation was more than 20%, which is too high to be explained by the core/shell tracks. The variation could be due to the geometrical effects of crater shape. Since we cannot reach the firm conclusion of the core/shell track formation, we describe both the possibilities but don't specify one of them.

Following sentences were added in page 11 of the manuscript. “Possible candidate is (i) geometrical effects of the crater shapes or (ii) core/shell tracks [40,41]. Further study is necessary to clarify the origin.”

R2-4b) The molecular dynamics (MD) simulations can be a good idea to understand the track formation in diamond. However, it could be challenging: In the cases of swift heavy ion irradiation, the ballistic atomic collision effects are negligible. The MD consider the energy deposition from electronic excitation only. However, in the case of MeV C_{60} irradiation, both the ballistic atomic collisions and the electronic excitation should be considered simultaneously. The electronic stopping S_e is higher than the nuclear stopping S_n , but S_n is not negligible. We will add them in the future plan. Thank you for the interesting suggestion.

C2-5) The authors refer on a number of occasions to the “velocity effect” and contend that this provides the explanation of the different keV/nm thresholds for C₆₀ and monoatomic ions. An explanation of the velocity effect can be found in their reference 21 and in references cited therein, but it would be useful to have a brief explanation of this (just a few lines) in the paper itself.

R2-5) Following sentences explaining the velocity effect was added in p. 3. “Higher velocity SHIs emit higher energy δ -electrons toward the radial directions, which excite the cylindrical region along the ion trajectory. The radius of the heated region is determined by the range of the δ -electrons. Consequently, the higher velocity SHIs form larger volume of the excited regions because of the longer δ -electron ranges, resulting in lower density excitation. Lower velocity ions result in higher density excitation, which is advantageous for track formation. We use the word “velocity effect” in this context [28,29].”

[28] Meftah, A. *et al.* Swift heavy ions in magnetic insulators: A damage-cross-section velocity effect. *Phys. Rev. B* **48**, 920-925; <https://doi.org/10.1103/PhysRevB.48.920> (1993).

[29] Dufour, C. & Toulemonde, M. Models for the description of track formation, in *Ion Beam Modification of Solids* Vol. 61 (eds Werner Wesch & Elke Wendler) Ch. 2, 63-104, (Springer, 2016).

C2-6) On a much more minor level:

Fig. 2(d) is probably superfluous – readers will understand the terms “crater” and “hillock” without this.

R2-6) We did not include Fig.2(d) in early draft. A co-author, who is a scientist of ion beam apparatus developments but not so familiar with ion-solid interaction, suggested us to include such a cartoon to help understanding the images of Fig. 2(a)-(c). It could be superfluous for the reviewer since the reviewer must be an authority and well knows about this field. The authors would like not to delete Fig. 2(d), if the reviewer allows it. If the reviewer strongly suggests us to delete Fig. 2(d), we will follow the suggestion.

C2-7) The statement on p5 that “the tracks observed from the surface normal direction are not perfectly circular but ellipsoids. This is probably due to the incident angle being 7° from the surface normal” cannot be correct given that the cosine of 7° is only 0.9925! And note that it should be “ellipse” rather than “ellipsoid”.

R2-7a) When the authors planned the irradiations, we had the same expectation as the

reviewer #2 described. However, we forgot the effect of the inclined sidewalls. As shown in Fig. R5, TEM observed not only the circular (elliptic) cross-section of the tracks but also the inclined sidewall of the track rods when the observation direction is inclined from the axial direction.

Since the track length under 9 MeV C₆₀ irradiation was $L \sim 50$ nm, $L \cdot \sin(7^\circ) = 0.122 \cdot L = 6$ nm. This contribution cannot be neglected since the mean track diameter was 7 nm. The following sentence was added in p.5.

“This is probably due to the ion incident angle being 7° from the surface normal, since the images of the track sidewalls overlapped with the track diameters. The contribution is roughly $L \cdot \sin 7^\circ \sim 6$ nm, which is comparable to the track diameter, where $L \sim 50$ nm, the mean track length of 9 MeV C₆₀ ions in diamond.”

R2-7b) The word “ellipsoid” was replaced by “ellipse”. Thank you.

Reviewer #3 (Remarks to the Author):

C3-1) The authors made an interesting observation of ion tracks induced by C₆₀ clusters in diamond. In general, ion tracks are easy to form in insulators but difficult to form in metals. However, diamond is one of the most unfavorable insulators to form tracks. The use of C₆₀ clusters or high energy heavy ions (up to GeV) would significantly increase the chance to induce tracks in materials that were previously considered unfavorable for track formation, including metals and metallic compounds (Henry et al., NIMB 67 (1992)

390-395; Barbu et al., NIMB 145 (1998) 354). For example, the authors have recently reported track formation by C₆₀ clusters in silicon (Amekura et al. Scientific Reports 11 (2021) 185). Although the irradiations by swift ions have been studied in diamond powders before (Zhang et al., NIMB 286 (2012) 262), ion tracks have never been observed in diamond. Thus, the observation of tracks in diamond by C₆₀ clusters has important implications for understanding the track formation mechanism, which is still controversial. However, based on the problematic HR-TEM image shown in Fig. 3, the authors interpreted the inside of the C₆₀ cluster induced tracks as the crystalline diamond. Consequently, the explanation of the high pressure of 0.1-10 GPa within the diamond during the track formation is also questionable.

R3-1a) Thank you for suggestive comments. We didn't know the paper by Zhang et al. on the diamond powder. We have added a sentence about this paper in the text.

However, please remind that Zhang et al. irradiated the powder to much higher fluences ranging from 5×10^{12} to 8×10^{13} ions/cm², i.e., 100-1600 times higher than the present fluence of 5×10^{10} ions/cm². We need caution when comparing with them.

R3-1b) The reviewer wrote "However, based on the problematic HR-TEM image shown in Fig. 3, the authors interpreted the inside of the C₆₀ cluster induced tracks as the crystalline diamond." We have irradiated a new sample and carried out the HR-TEM with more reliable conditions. Please read R3-2a).

R3-1c) The reviewer wrote "Consequently, the explanation of the high pressure of 0.1-10 GPa within the diamond during the track formation is also questionable." However, this is not the consequence from Fig. 3, but simply from the phase diagram (Fig. 5a), which was deleted when the present revision.

This is because that late Fig. 5a was a phase diagram under thermal equilibrium, but the ion track formation is processes under non-thermal equilibrium. Since the pressures of 0.1-10 GPa were the values under the thermal equilibrium, the authors gave up the description of the pressure.

In the i-TS calculations, for simplicity, the material is assumed to be heated up to very high temperatures with maintaining the atmospheric pressure. However, it is not the case for diamond: At the atmospheric pressure, the liquid phase does not exist in diamond. The only possible phase transition is the sublimation, which, however, requires much

higher energy of 8.35 eV/atom than the attainable energy by a 9 MeV C₆₀ ions (~3 eV/atom).

Nevertheless, we have observed the track formation under 9 MeV C₆₀ irradiation, indicating that another phase transition, i.e., the melting transition, is induced. However, the track formation is a non-equilibrium process. The equilibrium phase diagram cannot be directly applied. Since the experimental results are well explained by assuming the melting transition, the liquid phase is somehow stabilized under the track formation, i.e., a non-equilibrium condition.

The authors consider that one of the most possible processes is the pressurization and the stabilization of the liquid phase of diamond. However, under the non-equilibrium condition, the phase diagram can be deformed or possibly completely changed. Too much specification of the description “0.1-10 GPa” was deleted.

C3-2a) The HR-TEM image in Fig. 3 is unable to clearly show the atomic structure in the inside of the ion track. The lattices in Fig. 3 appear diffuse, and the boundary between the track and the surrounding material is unclear. The possibility of the amorphous components within the track cannot be ruled out, as they are more difficult to determine than the crystalline components within the track, particularly when the quality of this image is poor. I understand that taking a high quality of HR-TEM image in diamond is not an easy task, as the track formation in this material is difficult, and an extended milling time by Ga ions is required for the sample preparation. For better HR-TEM images of ion tracks, the authors should satisfy some important requirements, including but not limited to, the very thin sample thickness, the minimum radiation damage during sample preparation, and the good alignment among ion tracks, zone axis and electron beams.

The authors failed to align ion tracks with the zone axis of the sample due to the use of the polycrystalline diamond and the 7° ion-incident angle from surface normal. As shown in previous high-quality HR-TEM studies (Vetter et al., 141 (1998) 747; Sachan et al., JMR volume 32 (2017) 928), the three directions, i.e., the electron beams, the incident ions, and a low index zone axis of the material should be strictly aligned to show sharp lattices and a clear track boundary with the matrix. As the investigated material is polycrystalline diamond in this study, however, it is practically impossible to align the ions along a known, low index zone axis prior to ion irradiations. During the TEM observation, it is also impossible to align ion tracks along the electron beams by tilting the zone axis of the sample, thus resulting in an elongated ion track, rather than a circular ion track, projected on the top surface.

R3-2a) Thank you for advice.

- (i) We have thinned down some samples. Accidentally (luckily) one of them has an orientation $[3\ 2\ 1]$ without any tilting.
- (ii) The specimen was irradiated at the incidence of the surface normal direction with 9 MeV C_{60} ions. Therefore, the ion beam was introduced perpendicular to the specimen surface, i.e., parallel to $[3\ 2\ 1]$.

- (iii) The HR-TEM observation was carried out perpendicular to the specimen's surface, i.e., parallel to the tracks. Mis-observation of the track sidewalls as the track interior

Fig. R6. High-resolution transmission electron microscopy (HR-TEM) images of an ion tracks in diamond, formed under 9 MeV C₆₀ ion irradiation to the surface normal. The fluence was 5×10^{10} C₆₀/cm². The sample surface was accidentally aligned to the [3 2 1] orientation of diamond without any tilting. The sample was thinned down to ~36 nm thick by FIB milling, which is thinner than the track lengths of 52.1 ± 7.3 nm, excluding the contribution of unirradiated layers deeper than the track ends. While the Scherzer defocus $df_{sch} = - (4/3 C_s \lambda)^{1/2}$ was -57.9 nm for JEM-2100F (the spherical aberration coefficient C_s of 1.0 mm and the electron wavelength $\lambda = 2.5079$ pm), the defocus of -84 nm was applied for better images. **(a)**: HR-TEM image of four tracks. It is not clearly shown in the image, but the lattice fringes are recorded all over the image. **(b, c)**: Magnified images of two different ion tracks. Whole the regions including the inside the tracks show the lattice fringes. The fast Fourier transformed (FFT) images of **(d)**' the region on a track and of **(e)**' the region without tracks are shown in **(d)** and **(e)**, respectively. Line profiles of the FFT images along the diagonal lines shown in **(d, e)** are plotted in **(f)** under a relative unit. Except a sharp central direct spot, both the profiles were almost identical with each other. The calculated lattice fringes by the multi-slice method are shown in **(g)**. The experimental fringe period $d_{1-1-1} = 0.21 \pm 0.02$ nm as shown in **(b)**, which matches to the literature value $d_{1-1-1} = 0.2059$ nm.

is excluded.

- (iv) The sample was thinned down to 36 nm in thickness, which is thinner than the mean track length of 52.1 ± 7.3 nm. Most (more than $2SD = 95\%$) of tracks reach the other side of the specimen. In case, unirradiated deeper parts of the tracks do not exist.
- (v) We have observed more than twelve tracks. All the tracks show the lattice fringes inside.

Therefore, the lattice fringes in tracks are not spurious but real. Please see Fig. R6 and the caption.

*** The tracks are not in perfect circles but deformed. We consider that the deformed shapes are due to highly efficient partial recrystallization of tracks in group-IV semiconductors. The highly efficient recrystallization was evident in silicon in our recent work [Physica Scripta **98**, 045701 (2023)].

C3-2b) The authors claimed that the inside of the ion track is damaged crystalline diamond, rather than graphite or amorphous carbon (Fig. 3). However, the sample was

milled by extremely high fluence of Ga ions, which would induce significant radiation damage, more specifically the formation of amorphous carbon throughout the surfaces of the TEM sample. As a result, the Ga ions induced amorphous carbon on the surfaces would interfere the identification of amorphous carbon inside the track. As the amorphous components are more difficult to identify than the crystalline components directly from the HR-TEM image, the reviewer obtained the Fast Fourier Transform (FFT) images from two regions in the HR-TEM image of Fig. 3, that is, the center of the track (red frame) and a distance from the track (blue frame). As shown in Fig. R7, the diffuse ring in the center of the FFT image (bottom right) indicates more amorphous components within the track, while the absence of a diffuse ring in the center of the FFT

image (upper right) indicates less amorphous components at a distance from the track. This distribution of amorphous component as obtained by FFT is against the authors' claim that the inside of the ion is crystalline.

R3-2b) We have realized that it is not easy to exclude the existence of amorphous phase in the track even we observed the lattice fringes inside. Both the phases could exist with each other in the track. Furthermore, our previous observation conditions (the ion incidence of 7° which was not parallel to the zone axis, and the sample thickness thicker than the track length) cannot exclude the lattice fringes from the sidewalls of the tracks or deeper unirradiated parts of the tracks. Therefore, as shown in Fig. R6, we have fabricated new samples (much thinner than the mean track length, the surface normal matching to a crystallographic axis, and normal incidence of the ions). The results are shown in Fig. R6.

An FFT image over a track (Fig. R6d) and that from the region without tracks (Fig. R6e) were observed. Line-profiles of both the images are compared in Fig. R6f. Both the profiles were comparable except the central spot. Therefore, we consider that the amorphous signal in the tracks is comparable to that from unirradiated area. The main origin of the amorphous components could be due to the Ga thinning.

Anyway, lattice fringes in the tracks and amorphous signal independent from the tracks are consistent with our picture of damaged crystalline tracks (or partial recrystallization of tracks). At first, amorphous tracks are formed but partially recrystallized to diamond, instead of graphite, due to the highly efficient recrystallization in group IV semiconductors.

C3-2c) The formation of graphite-like carbon in the ion irradiated diamond powders was evidenced by the previous study of swift ion irradiation experiments, which clearly show a radiation-induced transformation to the graphite-like carbon by using electron diffraction, EELS, and Raman (Zhang et al., NIMB 286 (2012) 262). Raman is reliable to determine the graphite-like carbon, as the Raman signal is collected from many diamond grains, rather than a single grain in this study. The ring of graphite G(002) is also clearly identified from the electron diffraction from the diamond powders, for which the radiation damage is absent during the sample preparation. These results are contradicted with the interpretation of crystalline diamond in this study.

R3-2c) Zhang et al. irradiated the powder to much higher fluences ranging from 5×10^{12} to 8×10^{13} ions/cm², i.e., 100-1600 times higher than the present work of 5×10^{10} ions/cm². We need caution when comparing with these results.

Judging from the Raman spectra and EELS spectra reported by Zhang et al., the authors expect that the graphite signal was below the detected limit if Zhang et al. irradiated 100-1600 times lower fluences.

As described in C3-m8 (later shown), it seems that the reviewer #3 confuses the 285 eV peak (only graphite contributes) and the 295 eV peak (both diamond and graphite contribute) in EELS spectra shown in Fig. 4b. Zhang et al. reported the 285 eV peak (graphite), indicating that much graphite is formed in diamond powder irradiated with 1.4 GeV U ions. However, our samples show only 295 eV peak (diamond and graphite) but almost no 285 eV peak (graphite), indicating that only negligible graphite is formed in our samples irradiated with 9 MeV C₆₀ ions. We have measured several tracks and observed that the graphite signal was always very weak.

Please remember that the authors observed ion tracks but Zhang et al. did not. Both Zhang et al. and the present authors have not always observed the same results since former didn't but latter observed the tracks. The irradiation effect of 1.4 GeV U ions and 9 MeV C₆₀ ions can be "different". It is not good to assume that much of graphite must be formed in our samples because graphite was formed in the samples of Zhang et al. In fact, Zhang et al. observed a strong graphite peak at 285 eV in EELS. However, we have observed almost no graphite (285 eV) peak in EELS (Fig. 4b). This is not a conclusion from one track. The authors measured EELS mapping of several tracks. The results were always almost similar to Fig. 4b, i.e., diamond phase was always dominant.

Raman spectroscopy is not hopeful for our samples, because the ion range of 9 MeV C₆₀ ions is ~50 nm while the thickness of the sample is 0.3 mm. It is almost impossible to detect the signal from such thin irradiated layer against the huge background from the thick unirradiated layer.

C3-2d) The sample appears too thick for HR-TEM image based on the diffuse lattices (Fig. 3). According to the Methods, the top view sample for HR-TEM was pre-thinned to ~100 nm before the ion irradiation. The Kikuchi lines (Fig. 2g) also suggest that a thick sample (at least 100 nm) was used for the electron diffraction pattern. A much thinner sample (20 nm or below) is usually required for a high-quality HR-TEM, although the authors may look for a region with a thinner thickness for better HR-TEM images. When the quality of the HR-TEM is not good, one should check if the sample is too thick.

However, there is no other information for the sample thickness of the HR-TEM in the text.

R3-2d) This comment is correct. We have thinned down one of the new samples down to 36 nm thick.

C3-3) Knowing that the orientations are unknown for the polycrystalline diamond, one may wonder why the authors avoided the ion channeling effect by choosing the 7° angle from surface normal. The authors have clarified that the large grain in Fig. S7 can be considered as a single crystal, although the diamond is polycrystalline. For this HRTEM image, the zone axis can be determined as the [110] direction of diamond (SI Fig. 1), which is the same as the zone axis [110] in the diffraction pattern in Fig. 2g. However, this grain for the HRTEM image (Fig. 3) should be a different grain from the one in Fig. 2g due to a large angle ($\sim 70^\circ$) between their (1-11) planes in Fig. 2g and SI Fig. 1.

Their choice of the 7° angle from surface normal can be validated only when all the grains are oriented along the same orientation (e.g., [110]).

R3-3) The description “The incident angle was set to 7° from the surface normal to avoid ion channeling.” was totally stupid. We revised as “The incident angle was set to 7° from the surface normal.”

HRTEM and SAED were recorded from different TEM apparatuses. Bright field images and SAED were recorded using JEM-2100, but HR-TEM was done using JEM-2100F. When a TEM ring (with a sample) is mounted to a holder, we cannot control the in-plane angle of the sample as same as before, because the TEM ring is circular and no mark. Some samples were observed a few – several times. Every time the in-plane angle could be different. Therefore, Fig. 2g was replaced by a clear one, while the in-plane angle was different from the previous one.

C3-4) The reviewer #3 think that the crystalline components observed from the top-view HR-TEM are not solely from the inside of the cluster induced track. The crystalline components can be originated partially from the surrounding crystalline diamond, and/or at deeper depths of the sample. The track length is only 52.1 nm for 9 MeV C_{60} clusters and 17.3 nm for 2 MeV clusters, which are significantly shorter than the nominal thickness for the top view samples ~ 100 nm. In addition, the track width becomes narrower with increasing depths. All these allow the signals from the undamaged

crystalline diamond at a deeper depth to project onto the crater-like tracks for the top-view HR-TEM images.

R3-4) To answer these comments, we have fabricated new samples and thinned down to 36 nm thick, as already described. The sample was aligned to [3 2 1] orientation without any tilting, and it was irradiated with 9 MeV C₆₀ ions parallel to the [3 2 1] axis.

We have irradiated the sample only with 9 MeV C₆₀ ions but not 2 MeV C₆₀ ions to avoid the shorter tracks than the sample thickness of 36 nm. The mean track length and the standard deviation (SD) of 9 MeV C₆₀ ions in diamond were 52.1 nm and 7.3 nm. The thickness of 36 nm is shorter than the (mean length) – 2*SD = 52.1 – 2*7.3 = 37.5 nm, i.e., the probability of less than 2.3% for the lower 2SD. Deeper unirradiated parts do not exist in this case.

While we have observed more than ten tracks, all the tracks show the lattice fringes. Since the observation was performed parallel to the ion beam, the central white parts with the lattice fringes cannot attribute to the sidewalls of the tracks.

The surface craters/hillocks observed in Fig. 2b are ascribed to the white cores and surrounding black structures shown in Fig. R6b and c. The craters/hillocks are separately observed in Fig. R6, which are excluded from the origin of the observed lattice fringes from the sidewalls.

Other comments:

C3-m1) P1L2: High energy ions or fullerene ions create tracks not only in insulators, but also in metals or metallic compounds.

R3-m1) We know tracks are formed in metals and metallic compounds. Please look at L1 in the abstract. We have written “Injecting ... ions ... into *solids* can create ... ion tracks.” In this context, “solids” include metal, semiconductors, insulators, and so on. We intended that tracks are formed in even metals. Then the next sentence, we wrote “Although these tracks form in many insulators”, i.e., tracks are easily formed in insulators. “However, one of the exceptions is diamond.” This was the story which the authors intended.

P1L2: The phrase “Although these tracks formed in many insulators” is revised as “... in many materials”.

C3-m2) P2L7: In most solids, the total length of fission tracks is about 20 micrometers, and the length for each fission fragment is around 10 micrometers. The current statement

is misleading. The readers may wonder if this length “more than micrometers” is the total length or the length along one direction.

R3-m2) Thank you very much. The phrase “(typically ... and more than μm in length)” was replaced by “(typically ... and around 10 μm in length for each fragment)”.

C3-m3) P2L15: The authors should inform readers that ion tracks can be created in metals and metallic compounds, although tracks are normally easier to form in insulators.

R3-m3) The sentence was revised as “Over the past few decades, formation of ion tracks by SHIs has been a central topic of research in ion–solid interactions in various materials including metals [5], metallic compounds[6], oxide superconductors [7], semiconductors [8], inorganic insulators [9], and polymers[10].” And these references were added.

Ref.

- [5] Henry, J., Barbu, A., Leridon, B., Lesueur, D. & Dunlop, A. Electron microscopy observations of titanium irradiated with GeV heavy ions. *Nuclear Instruments and Methods in Physics Research Section B: Beam Interactions with Materials and Atoms* **67**, 390-395; (1992).
- [6] Barbu, A., Dunlop, A., Hardouin Duparc, A., Jaskierowicz, G. & Lorenzelli, N. Microstructural modifications induced by swift ions in the NiTi intermetallic compound. *Nuclear Instruments and Methods in Physics Research Section B: Beam Interactions with Materials and Atoms* **145**, 354-372; (1998).
- [7] Bourgault, D., Hervieu, M., Bouffard, S., Groult, D. & Raveau, B. 3.5 GeV xenon ion irradiation effects in the superconducting oxide $\text{YBa}_2\text{Cu}_3\text{O}_{7-\delta}$ ($\delta \leq 0.1$): A HREM investigation. *Nuclear Instruments and Methods in Physics Research Section B: Beam Interactions with Materials and Atoms* **42**, 61-68; (1989).
- [8] Kamarou, A., Wesch, W., Wendler, E., Undisz, A. & Rettenmayr, M. Radiation damage formation in InP, InSb, GaAs, GaP, Ge, and Si due to fast ions. *Phys. Rev. B* **78**, 054111; (2008).
- [9] Meftah, A., Brisard, F., Costantini, J. M., Dooryhee, E., Hage-Ali, M., Hervieu, M., Stoquert, J. P., Studer, F. & Toulemonde, M. Track formation in SiO_2 quartz and the thermal-spike mechanism. *Phys. Rev. B* **49**, 12457; (1994).
- [10] Papaléo, R. M., Silva, M. R., Leal, R., Grande, P. L., Roth, M., Schattat, B. & Schiwietz, G. Direct Evidence for Projectile Charge-State Dependent Crater Formation Due to Fast Ions. *Phys. Rev. Lett.* **101**, 167601; (2008).

C3-m4) P5L10: The best evidence to confirm the ion tracks is to compare the ion fluence with the track density, rather than using the reversal Fresnel contrast.

R3-m4) Following sentence was inserted in p.5. “The track density was $5.8 \times 10^{10} \text{ cm}^{-2}$, which was nearly comparable to the ion fluence of $5 \times 10^{10} \text{ C}_{60}/\text{cm}^2$.”

C3-m5) P6L2-6: The side view image (Fig. 2c) is not good enough to show the two-step structures. The best evidence is to show hillocks or caves from a high-quality side view image, such as in the paper by Ishikawa et al., Nanotechnology 28 (2017) 445708.

R3-m5) Since the surface was covered by Pt layer, it is difficult to observe tinny structures on the sample surface. Ishikawa et al. prepared TEM specimens of relatively soft insulators such as CaF_2 or so on using the crushing method, which is not applicable to the hardest material diamond. Another side view image is shown in Fig. R8, also showing no clear two-step structures.

Fig. R8. An expanded image similar to Fig. 2c in the text.

C3-m6) P6L20: This is an orientation along $[110]$ of cubic diamond, rather than a $\{110\}$ plane. Fig. 2g also needs corrections for the orientation.

R3-m6) Thank you. Both the sentence and Fig. 2g were revised.

C3-m7) P7L10: The typical period is 0.206 nm for $\{111\}$ planes, rather than 0.113 nm. The lattices with a distance of 0.206 nm can be measured from the HR-TEM image. The

authors seem to confuse the concepts between planes and orientations. Refer to Fig. 2g, for which the zone axis is [110] direction rather than {1 1 0} planes.

R3-m7) Thank you. We have revised them. We are ashamed of confusion.

C3-m8) P8L10: A peak around 295 eV, which is on the left of the 306 eV peak, can be ascribed to graphite. Amorphous graphite may form during the Ga ion sample preparation, or during the C₆₀ irradiations.

R3-m8) It is not correct. A typical peak for graphite is at 285 eV, not at 295 eV. Both graphite and diamond contribute to the 295 eV. The 285 eV peak is ascribed to the transition from the 1s to the (unoccupied) π -orbitals, which is only observed in graphite. The 295 eV is ascribed to the transition from the 1s to the (unoccupied) σ -orbitals, which is observed in both graphite and diamond.

Since signal of the graphite (285 eV) peak is very weak, the amorphous graphite formed by Ga beam or C₆₀ irradiation is very low.

C3-m9) P8L20: The EELS signal is related to, but not equal to, the density of the matter, as the electron energy loss peak also reflects variations in certain peaks of graphite or diamond. To show the core-shell structure, the authors should use the HAADF-STEM technique to show the Z contrast (or density) change.

R3-m9) While we have tried HAADF-STEM with a proper camera length, images were too weak to find tracks. If we increase the camera length to 20 cm, we can see the tracks. But it is not a true HAADF-STEM mode and the image is not proportional to the density.

Since we observed relatively large variations of EELS intensity depending on the distance from the track core, we speculated the core/shell tracks as one of the possibilities. However, the intensity variation was more than 20%, which was too large to be explained the core/shell tracks. Also other reviewer suggests the geometrical effect of the crater chapes. Since we don't have firm evidence, we raise both the possibilities but not specify the origin. HR-TEM (Figs. R6b and c) suggests the geometrical effect.

C3-m10) P10L17: The calculation is inconsistent with the observation of amorphous graphite by Raman (Zhang et al., NIMB 286 (2012) 262). The formation of the amorphous graphite is not due to the D-G phase transition, but more likely due to the ion-irradiation induced formation of Frenkel pairs.

R3-m10) As described in R3-2c), Zhang's work cannot be directly compared with the present work, since Zhang et al. irradiated the powder 100-1600 times higher fluences. These sentences were added in "Chemical phase inside the tracks". "As already

described, Zhang et al. irradiated diamond powder with 1.4 GeV U ions; No track formation was observed, but the graphitization was detected [18]. While the present study did not observe the graphitization, the work by Zhang et al. cannot be directly compared with the present study because Zhang et al. irradiated the powder to much higher fluences ranging from 5×10^{12} to 8×10^{13} ions/cm², i.e., 100–1600 times higher than the present study of 5×10^{10} ions/cm².”

While the reviewer wrote “the ion-irradiation induced formation of Frenkel pairs”, does it mean the collisional (nuclear) formation of Frenkel pairs? While the authors (roughly) read the Zhang’s paper (NIMB286(2012)262), the authors did not understand why the graphitization is ascribed to the collisional Frenkel pair formation?

First of all, the Frenkel pair formation (point defect formation) does not result in the ion track formation. Therefore, our calculations aimed to reproduce the track formation in diamond, which SHOULD be INCONSISTENT with Zhang’s observations where the track formation was not observed.

Since the authors observed the track formation but not graphitization and Zhang et al. observed graphitization but not track formation, the phenomenon observed by the authors should be considered different from that observed by Zhang et al. The inconsistency between them is a natural consequence.

C3-m11) P11L17: Give references when the velocity effect was first introduced.

R3-m11) P4: Two references were added when the velocity effect was introduced.

Ref.

[27] Meftah, A. *et al.* Swift heavy ions in magnetic insulators: A damage-cross-section velocity effect. *Phys. Rev. B* **48**, 920-925; <https://doi.org/10.1103/PhysRevB.48.920> (1993).

[28] Dufour, C. & Toulemonde, M. Models for the description of track formation, in *Ion Beam Modification of Solids* Vol. 61 (eds Werner Wesch & Elke Wendler) Ch. 2, 63-104, <https://doi.org/10.1007/978-3-319-33561-2> (Springer, 2016).

C3-m12) P14L1: The Ga ion thinning would induce additional radiation damage in the cluster ion damaged diamond. The Ga ion induced damage is difficult to separate from the damage induced by the cluster C₆₀.

R3-m13) We know the problem of the Ga beam thinning damage. However, there is no thinning method instead. Mechanical thinning, e.g., crushing thinning, does not work, because this is the hardest material. As an attempt, we tried to cut a CVD-diamond thick film using a diamond-powder-blade. One day after, no scar was observed on the diamond

thick film, but we lost the blade thickness. Shock cutting may work. But the shock cutting often break the diamond samples to unintended shapes. Cutting and thinning of the diamond samples are not easy tasks.

C3-m13) P14L10: The hardness of diamond may be related to the difficulties of sample thinning by Ga ions, but the dominant reason is the lower sputtering coefficient of diamond as compared to that of silicon.

R3-m13) Yes, the direct reason is the lower sputtering yield of diamond as compared to that of silicon, as suggested by the reviewer. We speculated that there is a relationship between the lower sputtering yield and the hardness of diamond, but we cannot show the evidence at the moment. We follow the suggestion from the reviewer and change the sentence from “Because of the extreme hardness of diamond...” to “Because of much lower sputtering yield of diamond ...”.

Reviewer #4 (Remarks to the Author):

The manuscript describes ion track formation in CVD grown diamond films using C₆₀ cluster irradiation. So far ion tracks have not been observed in diamond using single ion irradiation and I am not aware of any previous experiments with cluster beams. The investigation follows a similar paper from the authors on ion tracks in silicon produced by C₆₀ clusters. Like in diamond, ion tracks had not been observed in Si following single ion irradiation.

The ion tracks were observed using TEM combined with EELS and calculations using a simple thermal spike model were performed to help explain the formation of ion tracks.

The main results from the experiments are:

- i. The formation of ion tracks using C₆₀ clusters with energies between 2 and 9 MeV (no tracks were observed for 1 MeV clusters)
- ii. A track formation threshold of 23.9 keV/nm from liner fit to track data was established
- iii. The material in the tracks is crystalline (yet defective), no details on the nature of the defects was provided
- iv. EELS is consistent with a core shell structure of the tracks.

C4-i). The main observation that ion tracks in diamond are formed using cluster beams

is indeed interesting to the ion track community, although not very surprising. The manuscript provides some suggestions of how this can happen (in particular as irradiation with single ions of higher energy loss does not lead to track formation) but these are only speculations not backed up by experimental results. These suggestions include the velocity effect or synergies between electronic and nuclear stopping, both mechanisms have been discussed in the context of track formation previously.

R4-i) The reviewer wrote “The manuscript provides some suggestions of how this can happen (in particular as irradiation with single ions of higher energy loss does not lead to

Fig. 5. (c) Inelastic thermal spike (i-TS) calculations of diamond irradiated with C_{60} ions: Changing S_e from 20 to 80 keV/nm, diameters of the maximum molten regions were calculated by the i-TS model with two values of latent heat of fusion L_f (red and blue curves) and two velocities (solid and broken curves). The experimental data of C_{60} were closely reproduced by the 0.05 MeV/u curves. The absence of track formation by 1.03 GeV Bi and by the highest S_e of U are indicated by a triangle and a square, respectively. They are consistent with the curves of 5 MeV/u. Error bars represent SD of experimental diameters of more than fifty tracks for each S_e .

track formation) but these are only speculations not backed up by experimental results.” Definitely the authors cannot agree with the reviewer’s statement “these are only speculations not backed up by experimental results.”

As the authors described that the synergy effect between S_e and S_n could contribute the track formation under MeV C_{60} ion irradiation, the description made confusion. (We deleted these descriptions from Introduction). We consider that both the appearance of tracks by MeV C_{60} ions and the non-appearance of the tracks by GeV U ions can be explainable by the i-TS calculations with the velocity effect as shown in Fig. 5c.

Solid and broken curves are track diameters calculated by the melting transition of the i-TS model, respectively, for low velocity (0.05 MeV/u) and high velocity (5 MeV/u). Red and blue curves using the latent heat of fusion $L_f = 62.5$ and 125 kJ/mol, respectively. At the low velocity, the solid curves reach the threshold of ~ 20 keV/nm and reproduce the experimental track diameters of 2– 9 MeV C_{60} ions shown by green circles. At the high velocity (5 MeV/u), the broken curves reach the threshold of ~ 50 keV/nm and are consistent with no track formation by 1.03 GeV Bi ions (purple triangle) and at the Bragg peak of GeV Au ions (purple square).

Therefore, we provide, in this manuscript, an answer to this long-unsolved question for diamond from combination of experimental results and the calculations as shown in Fig. 5c. *This is a conclusion from a combination with the experiments and calculations, but not from pure speculations.*

C4-ii) Often a threshold for the electronic energy loss for track formation is derived from ion track data and it has previously been argued that the threshold for some materials is higher than what can be obtained with single ions (thus track formation is not observed). In the current case I think the threshold value reported is not significant. Firstly clearly it is below the energy loss that has been achieved with single ions (where no tracks have been observed) and is thus only applicable for the particular cluster beam. Secondly it was derived from a simple linear fit to data that clearly looks not to be linear and thus will probably have a huge uncertainty (which has not been reported). The value is thus not very meaningful.

R4-ii) The reviewer suggests the possible large uncertainty of the threshold value. Since it is reasonable, we use the threshold value of 24 keV/nm instead of 23.9 keV/nm.

Since the tracks was not observed under 1 MeV C₆₀ irradiation ($S_e = 21$ keV/nm) but observed under 2 MeV C₆₀ irradiation ($S_e = 29$ keV/nm), the threshold should be between them.

Since this is the first time to determine the threshold S_e of diamond experimentally, it is rather meaningful even including large error.

C4-iii) The nature of the defects is not specified but it appears that the track is consistent with a trail of defects. It may be difficult to provide more details on the nature of the defects but for an attempt on a mechanistic insight into track formation that would be important.

R4-iii) Thank you for important suggestion. We will try to clarify the nature of the defects in future works.

C4-iv) The core shell structure is also not further explained. Could that be a surface effect resulting from the craters formed given the TEM samples are very thin? There is no indication of why this core shell structure is similar to what has been observed previously in (to my knowledge) amorphous materials.

R4-iv) Since we observed variation of EELS intensity for the center of the track to faraway from the core and the variation was similar to the core-shell track, we have just speculated it. However, the EELS intensity variation could be due to the geometrical effect of crater shapes. Since we cannot reach the firm conclusion of the core/shell track formation, we describe both the possibilities but don't specify one of them.

As shown in Figs. 3b and 3c, HR-TEM clarified that a track has a white core and black periphery. However, the periphery does not distribute circularly. At the moment, the authors cannot judge whether the tracks are core-shell type or with craters.

Following sentences were added in page 9 of the manuscript. "Possible candidate is (i) geometrical effects of the crater shapes or (ii) core/shell tracks [40,41]. Further study is necessary to clarify the origin."

C4-v-1) The thermal spike calculations use two simple heat diffusion equations with equilibrium values for the specific heat and thermal conductivities. While this model has indeed be used prolifically to describe aspects of track formation, it is a gross oversimplification and has been heavily criticised (for example see S. Klaumunzer, Mat Fys Medd Dan Vid Selsk 52, 293 (2006)). Track formation is highly non equilibrium process that occurs on the nanoscale and the use of equilibrium quantities and phase

diagrams not justified. Its success is mostly driven by its simplicity and the moderately good agreement with experimental track radii (which is not surprising in many cases as the electron-phonon coupling is often treated as a fitting parameter). While I agree that some interpretations of the model are useful to get insights in the track formation process, it cannot and should not be used to explain how tracks form. This is exactly what the authors do and my main criticism.

R4-v-1) The corresponding author read the paper written by Klaumunzer (2006) and has realized that i-TS has many problems. I have learned a lot. Ok, we give up using the equilibrium phase diagram (old Fig. 5a) to discuss a non-equilibrium process of the ion track formation.

In our calculations, the electron-phonon mean-free-path λ was determined from the bandgap energy of diamond, which was not used as a fitting parameter.

C4-v-2) For this reason, molecular dynamics simulations have superseded the i-TS model in more recent publications on mechanistic insights into ion track formation. The fundamental assumption of the authors is that either melting, sublimation or a transition from diamond to graphite need to occur (the model requires a local phase transition to define the track boundaries). There is no evidence of any phase transformation in the data, only defect formation. Thus, the entire interpretation that high pressure is generated and

Fig. R8. An expanded image similar to Fig. 2c in the text.

required for track formation is not substantiated.

R4-v) Thank you for good suggestion. In future work, the authors would like to apply the molecular dynamics simulations to several MeV C_{60} irradiation to diamond. However, it is not easy since C_{60} ions are in low energies (from 2 MeV to 9 MeV C_{60} , i.e., from 30 keV/C to 150 keV/C), we need to consider both the ballistic atomic collisions and electronic excitation at the same time. Some developments in the two-temperature MD simulation methods are necessary.

While the reviewer claimed that no evidence of the phase transition, look at again Fig.R8. You see the damage regions of quite high aspect ratios. Are there any reported mechanisms except the ion track formation to explain such high aspect ratios? If they are due to the nuclear energy loss, they must be much more circular as collision cascades. Without the phase transition along the high energy ions, how can they be explained? A phase transition to an amorphous phase is induced, but the amorphous phase is soon recrystallized. There are many reports on the ion tracks which consist of not amorphous but damaged crystals, such as CaF_2 , CeO_2 , UO_2 , etc.

C4-vi) In summary, while the general observation that ion tracks form in diamond upon cluster irradiation is certainly interesting to the ion track community, the manuscript does not provide any new insights into track formation nor can it explain the experimental observations other than using well discussed qualitative notion of the ion velocity and synergy between electronic and nuclear stopping.

R4-vi) The authors do not agree with this conclusion, and would like to stress that this paper has clarified the long-unclarified question in a well-known material diamond, i.e., why ion tracks are NOT formed under high velocity monoatomic ions, e.g., GeV U ions. This paper has presented a new finding that the tracks are formed in diamond under 2–9 MeV C_{60} irradiation. The i-TS calculations successfully explained both the behaviors, i.e., no track formation under GeV monoatomic ions and the track formation under MeV C_{60} ions, only changing the ion velocity.

Another important finding in the present work is that our results experimentally confirmed that the ion tracks are formed even in diamond, if we use the cluster ions. *Furthermore, the ion tracks are formed in diamond only by the acceleration energy of 2 MeV, i.e., the terminal voltage of 1 MV for tandem accelerators.* It is quite enough low voltage for considering industrial applications.

The ion tracks formed in diamond can be used to form semiconductor circuits in a new semiconductor diamond having much more excellent semiconductor properties. The track formation could be applied to cut the hardest material diamond for industrial

applications. Recently, quantum information devices (e.g., NV-centers) in diamond received much attention. Ion beams are one of the most hopeful methods for fabricating the quantum devices in diamond. The track formation in diamond is not far away from these topics.

Anyway, ion tracks are formed in diamond with very low energy of 2 MeV, if C_{60} ions were used. This low energy is very attractive for industrial applications.

Reviewers' Comments:

Reviewer #2 (Remarks to the Author):

The authors have provided detailed and satisfactory responses to all of the issues that I raised in my original review and where necessary have amended the manuscript.

I suggested in my review that Fig. 2(d) was arguably superfluous but the authors have indicated in their response that they would like to retain this figure. I think this is an editorial decision but have no strong objections to keeping the figure.

As all of my concerns have been addressed, I have no further issues to raise and believe that this paper merits publication.

Reviewer #3 (Remarks to the Author):

In the revised manuscript, the authors insist on their explanation of crystalline diamonds in the track core. In the last review, I disputed this explanation mainly because of the interference of the unirradiated crystalline sample (i.e., the much thicker sample thickness than the track length and the possible tapering along a track), the diffuse ring from the amorphous matter on the ion track in the FFT image, and the poor quality of the TEM/STEM images. Although the authors have attempted to improve the HRTEM image with a single thinner sample, the quality of the HRTEM image remains poor in the revised manuscript as they chose a high-index zone axis, i.e., [321], rather than a low-index zone axis, e.g., [110]. They have not attempted to improve other TEM/STEM/EELS images for which the sample thicknesses remain too thick. Thus, details of the track morphology and defect distribution along the track remain unclear. The authors have not mentioned (or realized) that the FFT image (in Fig. 3c) clearly shows a ring of graphite (002) on the track core, indicating that the main conclusion on the crystalline diamond in the track core is incorrect. Without the details of the track nature, they have failed to provide a mechanism to explain how the C60 irradiations produce such track morphology or defect distribution in diamonds. In addition, the paper lacks novelty as the authors reported a similar result in silicon before. Therefore, I recommend rejecting the publication of this paper in a high-ranking journal like Nature Communications.

Detailed comments:

The sample thickness should be less than the track length to obtain reliable structural information on the track and avoid the interface of the unirradiated matrix. All the Z contrast/HRTEM/EELS images should have a suitable sample thickness. In addition, a high-quality HRTEM requires a low-indexed orientation to better show lattices. However, the authors have only improved the sample thickness for the HRTEM image (Fig. 3) for a single sample, and this sample orients along a strange, high index zone axis of [321], showing only one set of R vectors of diamond (-111). Without careful calibration, one cannot readily

determine the structure of the sample based on the measured lattice spacing of 0.22 nm, which is not so close to the theoretical one (0.203 nm). A reliable HRTEM and FFT analysis requires at least two sets of R vectors to identify the structure by the angle and ratio between two R vectors in the FFT image. The authors also should clarify if the electron irradiation during the HRTEM imaging alters the morphology, forming the lattice on top of the track core. In addition, a comparison of Fig. 3d and Fig. 3e demonstrates a diffuse ring with two spots pointed to the direction of 2 or 8 o'clock. This ring and the two spots belong to graphite (002) based on its ratio with diamond (-111), which is direct evidence that the amorphous graphite does exist in the track core.

There are no improvements on the side-view (Fig. 2C) and EELS (Fig. 4) images. Although the authors claimed a two-step structure, i.e., a large crater on the top followed by a cylindrical track, this structure is hard to identify from Fig. 2c, which remains unclear. To validate their two-step model, the authors should exclude the possibility of the track tapering, i.e., a larger track width on the top surface followed by a gradually narrowing track or even the disappearance of any damaged trail at deeper depths but significantly shorter than the theoretical ion range, as often seen in some materials (e.g., CeO₂ and TiO₂) where ion tracks are difficult to form. A high-quality side-view image can better show the tapering structure, with the damage disappearing at deeper depths. The authors should present more representative side-view TEM images to validate their model. If the tracks are really tapering, rather than a cylinder as the authors claimed, we should review the reliability of the measurements of track widths (Fig. 2) and the fit of their inelastic thermal spike (i-TS) calculations (Fig. 5). In addition, a top view HRTEM image of a tapering track may still show lattices on the track top, as the signals of the lattices may come from the sidewall of the gradually narrowing ion track.

The authors judge the crystalline track core by the lattices on the top of the ion tracks in Fig. 3. There are many possibilities to observe lattices on the ion track. The unclear TEM/HRTEM images do not provide reliable information to reveal the details of the track nature. One may wonder what the details of the track nature are. Is the track core a single diamond with the same orientations as the crystalline matrix? Does the diamond in the track core break up into smaller crystalline diamond grains oriented along the same direction? Does the track core consist of amorphous matter and crystalline diamond closely beneath the upper surface? The track becomes weak and disappears at deeper depths where the amorphous material readily returns to the crystalline diamond during track formation. We can still observe lattices on the ion track that consists of the amorphous material and crystalline diamond throughout the entire thickness.

The peak at 285 eV appears strong in the EELS analysis of GeV ion-induced tracks in diamonds (Zhang et al. 2012). The same 285 eV peak is weaker but recognizable in Fig. 4 in the current manuscript. The authors should not emphasize the differences but ignore their similarities. The authors have shown the four peaks from 292 to 328 eV without including the more critical peak of 285 eV. The authors may improve the EELS image and detect more significant graphite signals by either using a thinner sample or presenting the intensities of the peak of 285 eV against the distance from the track center.

Reviewer #4 (Remarks to the Author):

The authors have addressed most of the minor comments but my main point has not been addressed. I think there is a fundamental misunderstanding. The authors clearly state that a phase transformation is required for track formation. This is in my view not the case. An ion track is a long columnar defect structure (as the authors rightly show in Fig R8). The energy of the ions is coupled into the atomic structure along the trajectory, largely without generating atomic collisions. This can lead to the formation of long defect structures but does not necessarily require a phase transformation. At this scale and given the high non equilibrium nature of the process the simple phase considerations from temperatures calculated by the thermal spike model, which makes use of equilibrium properties, has always been questionable. That's why it has been superseded by MD simulations which can better describe the processes atomistically. Sure, the pure discovery of the ion tracks in diamond is interesting but I would expect for the study to be published in Nature Communications, that there needs to be some solid evidence about the track formation process based on state of the art modelling techniques. Unfortunately I don't see this in the manuscript. My suggestion would be to perform molecular dynamics simulations before publication, which may bring the required insights. Together with the experimental results, this would make a nice contribution to the journal.

One-to-one Response to Reviewers' comments

We express our gratitude to the editor and the reviewers for generous allocation of time and their valuable input in the form of insightful comments and suggestions.

Reviewer #2:

C2-1) The authors have provided detailed and satisfactory responses to all of the issues that I raised in my original review and where necessary have amended the manuscript.

C2-2) I suggested in my review that Fig. 2(d) was arguably superfluous but the authors have indicated in their response that they would like to retain this figure. I think this is an editorial decision but have no strong objections to keeping the figure.

R2-2) Thank you for a positive response for retaining Fig. 2(d), which was renumbered to new Fig. 3(a) in the revised manuscript. This figure could be superfluous for well-experienced readers, but possibly helps understanding of less-experienced readers.

C2-3) As all of my concerns have been addressed, I have no further issues to raise and believe that this paper merits publication.

R2-3) Thank you for agreeing to publication of this paper.

Reviewer #3:

R3-0) Before answering each comment from Reviewer #3, the authors briefly describe major improvements in experimental results (i)-(v), which would make the following one-to-one responses more comprehensive. Some of the new results have been also already predicted by Reviewer #3:

First of all, we have purchased (i) single crystalline diamond samples with (100) face. Now we were easily able to inject (ii) the C_{60} beams to diamond parallel to a [100] zone axis. Simultaneously, (iii) STEM observations were carried out from just the direction of the [100] axis. Of course, the TEM samples were fabricated to (iv) thinner than the previous samples, much thinner than the mean track length, which guaranteed that almost all the tracks reach to the other side of the sample. While the conventional HRTEM (i.e., not scanning) was used in the previous observations, (v) the Scanning TEM(STEM) was used this time, which drastically reduced artifacts. (Strange images observed in the previous manuscripts were probably due to artifacts.)

Consequently, two important results were obtained: (I) a certain size of amorphous region without lattice fringes was observed in ion tracks by HR-STEM. (II) STEM/EELS line-scan detected clear graphite signal (π -bonds) in ion tracks. Therefore, the authors

Fig. RR1. STEM-BF images of ion tracks formed under 9 MeV C_{60} ion irradiation incident to the [100] zone axis of a single crystalline diamond. The ion fluence was $5 \times 10^{10} C_{60}/cm^2$. (a) low and (b) medium magnification images. (c) a fast-Fourier transform (FFT) image of high magnification image (not shown). (d) was reconstructed from signal inside the red circle in (c) by filtering and the inverse FFT. The sample was thinned down to ~ 36 nm thick by FIB milling, which is thinner than the mean track length of 52.1 ± 7.3 nm, excluding the contribution of unirradiated bottom deeper than the track ends.

have realized that the suggestions from Reviewer #3 were correct. We gave up the idea of damaged crystalline ion tracks in diamond. Again, we appreciate Reviewer #3 for kind suggestions.

Fig. RR2. STEM-EELS line-scan from the center of an ion track to unirradiated region. (a) a STEM image of ion tracks observed under conditions optimized for EELS measurements but not for STEM. The purpose of this low-quality image (Fig. 5a) is to indicate the location of the line-scanning of STEM-EELS was performed along a green line. The EELS spectra of the C-edge at various distance from the track core are shown in (b) with the standard data of diamond and graphite. The π^* peak intensity was plotted against the distance as shown in (c), which decreases with the distance. A horizontal broken line is guide for the eye.

C3-1) In the revised manuscript, the authors insist on their explanation of crystalline diamonds in the track core. In the last review, I disputed this explanation mainly because of the interference of the unirradiated crystalline sample (i.e., the much thicker sample thickness than the track length and the possible tapering along a track), the diffuse ring from the amorphous matter on the ion track in the FFT image, and the poor quality of the TEM/STEM images. Although the authors have attempted to improve the HRTEM image with a single thinner sample, the quality of the HRTEM image remains poor in the revised manuscript as they chose a high-index zone axis, i.e., [321], rather than a low-index zone

axis, e.g., [110]. They have not attempted to improve other TEM/STEM/EELS images for which the sample thicknesses remain too thick. Thus, details of the track morphology and defect distribution along the track remain unclear. The authors have not mentioned (or realized) that the FFT image (in Fig. 3c) clearly shows a ring of graphite (002) on the track core, indicating that the main conclusion on the crystalline diamond in the track core is incorrect. Without the details of the track nature, they have failed to provide a mechanism to explain how the C60 irradiations produce such track morphology or defect distribution in diamonds. In addition, the paper lacks novelty as the authors reported a similar result in silicon before. Therefore, I recommend rejecting the publication of this paper in a high-ranking journal like Nature Communications.

A3-1) We deeply apologize the confusion. Following the suggestions from the Reviewer #3 written in the first review, observations were carried out more well-defined conditions with purchasing single crystalline samples. Now we were able to carry out ion irradiation and TEM observation from the same crystallographic direction. Also thinner samples than before were used. Now we observe no lattice fringes in the tracks by STEM and graphite-like signal by EELS. The authors have given up the idea of the crystalline tracks in diamond.

Detailed comments:

C3-2) The sample thickness should be less than the track length to obtain reliable structural information on the track and avoid the interface of the unirradiated matrix. All the Z contrast/HRTEM/EELS images should have a suitable sample thickness. In addition, a high-quality HRTEM requires a low-indexed orientation to better show lattices. However, the authors have only improved the sample thickness for the HRTEM image (Fig. 3) for a single sample, and this sample orients along a strange, high index zone axis of [321], showing only one set of R vectors of diamond (-111). Without careful calibration, one cannot readily determine the structure of the sample based on the measured lattice spacing of 0.22 nm, which is not so close to the theoretical one (0.203 nm). A reliable HRTEM and FFT analysis requires at least two sets of R vectors to identify the structure by the angle and ratio between two R vectors in the FFT image. The authors also should clarify if the electron irradiation during the HRTEM imaging alters the morphology, forming the lattice on top of the track core. In addition, a comparison of Fig. 3d and Fig. 3e demonstrates a diffuse ring with two spots pointed to the direction of 2 or 8 o'clock. This ring and the two spots belong to graphite (002) based on its ratio with diamond (-111), which is direct evidence that the amorphous graphite does exist in the track core.

A3-2) Thank you. We have realized difficulty using sample with high-index zone axis. At the present time, a zone axis of [100], i.e., normal incidence to one of the crystal surfaces, was applied. Then then authors have observed amorphous region in track as shown in Fig. RR1.

C3-3) There are no improvements on the side-view (Fig. 2C) and EELS (Fig. 4) images. Although the authors claimed a two-step structure, i.e., a large crater on the top followed by a cylindrical track, this structure is hard to identify from Fig. 2c, which remains unclear. To validate their two-step model, the authors should exclude the possibility of the track tapering, i.e., a larger track width on the top surface followed by a gradually narrowing track or even the disappearance of any damaged trail at deeper depths but significantly shorter than the theoretical ion range, as often seen in some materials (e.g., CeO₂ and TiO₂) where ion tracks are difficult to form. A high-quality side-view image can better show the tapering structure, with the damage disappearing at deeper depths. The authors should present more representative side-view TEM images to validate their model. If the tracks are really tapering, rather than a cylinder as the authors claimed, we should review the reliability of the measurements of track widths (Fig. 2) and the fit of their inelastic thermal spike (i-TS) calculations (Fig. 5). In addition, a top view HRTEM image of a tapering track may still show lattices on the track top, as the signals of the lattices may come from the sidewall of the gradually narrowing ion track.

A3-3) It is still difficult for us to have a good image of side-view. In the top-view, the

track thickness from the surface to the tail of the track, typically 50 nm thick, contributes the image. However, in the side-view, the track thickness corresponding to the track diameter, i.e., typically only 5 nm thick, contributes. This different contributing thickness is one of the origins of weak contrast.

Furthermore, it seems that long illumination by e-beam induces recovery of the tracks. Now the side-view sample show very faint images only.

While Fig. RR3(right) is not clear enough, this is one of the best images we obtained. This figure shows no carrot-shaped tracks (i.e., tracks whose diameters decrease with depth), which Dunlop et al. observed in YIG crystals [1]. These observations indicate that the track-heads with wider diameters exist close to the surface only, suggesting that they may be in the form of either surface-craters or hillocks. Since the inclined view from the top side (Fig. RR3(left)) shows the clearly wider track heads and the thinner track bodies, this observation is difficult to explain the track tapering effect with depth. However, further studies in future are necessary to clarify the two-step structures.

[1] A. Dunlop et al., Nucl. Instr. Meth. B 132, 93 (1997).

C3-4) The authors judge the crystalline track core by the lattices on the top of the ion tracks in Fig. 3. There are many possibilities to observe lattices on the ion track. The unclear TEM/HRTEM images do not provide reliable information to reveal the details of the track nature. One may wonder what the details of the track nature are. Is the track core a single diamond with the same orientations as the crystalline matrix? Does the diamond in the track core break up into smaller crystalline diamond grains oriented along the same direction? Does the track core consist of amorphous matter and crystalline diamond closely beneath the upper surface? The track becomes weak and disappears at deeper depths where the amorphous material readily returns to the crystalline diamond during track formation. We can still observe lattices on the ion track that consists of the amorphous material and crystalline diamond throughout the entire thickness.

A3-4) Many thanks for the Reviewer #3. We have finally found amorphous track cores in ion tracks as shown in Fig. RR1. An amorphous track-core in white is separated by black shell regions from unirradiated matrix. Since this is a STEM-BF image, the white track-core has lower density or thinner thickness. The black shell region could have higher density, which could be pressurized during the track formation processes. Further details of the track structures will be studied in future work.

C3-5) The peak at 285 eV appears strong in the EELS analysis of GeV ion-induced tracks

in diamonds (Zhang et al. 2012). The same 285 eV peak is weaker but recognizable in Fig. 4 in the current manuscript. The authors should not emphasize the differences but ignore their similarities. The authors have shown the four peaks from 292 to 328 eV without including the more critical peak of 285 eV. The authors may improve the EELS image and detect more significant graphite signals by either using a thinner sample or presenting the intensities of the peak of 285 eV against the distance from the track center.

A3-5) The authors appreciate Reviewer #3 for valuable suggestion. Using a thinner sample, the authors have detected the 285 eV peak (sp²) more clearly in EELS spectra. This observation is consistent with the broad FFT image due to (amorphous) graphite as shown in Fig. RR1(c). According to EELS line-scanning, the 285 eV peak (graphite) intensity decreases with the distance from the track center as shown in Fig. RR2(c).

**To respond the reviewer #3's comments and to describe the revised experimental results, many parts in the Result section in the manuscript have been rewritten. The revised parts are indicated in a separate file "Ms-2R9_highlighted.docx".

** Figure RR3(Right) was added to old Fig. 2 panel (in the main manuscript). Consequently, Fig. 2 had eight images in a figure panel, which we considered too many in a panel. Then, we divide the old Fig. 2 into two panels, i.e., new Fig. 2 and new Fig. 3 in the revised manuscript. Old Figs. 3, 4, and 5 were renumbered to new Figs. 4, 5, and 6.

Reviewer #4 (Remarks to the Author):

C4-1) The authors have addressed most of the minor comments, but my main point has not been addressed. I think there is a fundamental misunderstanding. The authors clearly state that a phase transformation is required for track formation. This is in my view not the case. An ion track is a long columnar defect structure (as the authors rightly show in Fig R8). The energy of the ions is coupled into the atomic structure along the trajectory, largely without generating atomic collisions. This can lead to the formation of long defect structures but does not necessarily require a phase transformation. At this scale and given the high non equilibrium nature of the process the simple phase considerations from temperatures calculated by the thermal spike model, which makes use of equilibrium properties, has always been questionable. That's why it has been superseded by MD simulations which can better describe the processes atomistically. Sure, the pure

discovery of the ion tracks in diamond is interesting but I would expect for the study to be published in Nature Communications, that there needs to be some solid evidence about the track formation process based on state of the art modelling techniques. Unfortunately, I don't see this in the manuscript. My suggestion would be to perform molecular dynamics simulations before publication, which may bring the required insights. Together with the experimental results, this would make a nice contribution to the journal.

A4-1) Thank you for an important suggestion. We have carried out the two-temperature MD simulations. In the simulations, a several MeV C_{60} ion was modelled as a projectile with high S_e but slow velocity, and was compared with a few GeV heavy monoatomic ion having high S_e and high velocity. This model of C_{60} ion is a quite simple model, which could be improved in future study. The calculated results are shown in Fig. RR4:

The simulations were carried out at two velocities of 0.05 MeV/u and 4.93 MeV/u and at three S_e of 20, 40, and 60 keV/nm, i.e., totally six conditions. At the high velocity of 4.93 MeV/u, a track was not formed at 20 keV/nm, and only a discontinuous track is formed at 40 keV/nm. While a track was formed at 60 keV/nm. The experimental confirmation is difficult, because the Bragg peak maximum of U ion in diamond provides S_e of 50 keV/nm only.

In the case of the low velocity of 0.05 MeV/u, always larger tracks than those at the high velocity were calculated. S_e dependence of track diameter for experiments and calculations are summarized in Fig. RR4(c). The experimental results of C_{60} ion irradiations are shown by green circles, which are well reproduced by the TT-MD calculations with the low velocity of 0.05 MeV/u (red cricles). The no track formation of diamond with swift heavy monoatomic irradiations are almost reproduced. (See purple symbols and cyan squares.) **Whole the parts of Discussion section have been revised. The revised parts are indicated in a separate file "Ms-2R9_highlighted.docx".**

(a) 4.93 MeV/u

(b) 0.05 MeV/u

Fig. RR4. Two-temperature molecular dynamics simulations of ion track formation in diamond. Top- and side-views of simulation cells are shown for (a) high (4.93 MeV/u) and (b) low (0.05 MeV/u) velocity irradiations, at three different $S_e = 20, 40,$ and 60 keV/nm. The simulations were performed up to 100 ps after the ion impact. The simulated and experimental results on the track diameters were plotted in (c). Green circles represent simulation results for low velocity (0.05 MeV/u), which well reproduced the experimental track diameters under C_{60} ion irradiations between 2 and 9 MeV. The absence of track formation by 1.03 GeV Bi ions and by the highest S_e of U ions are indicated by a triangle and a square, respectively. The simulated diameters for the high velocity (4.93 MeV/u) are shown by cyan squares, which almost reproduced the experimental results of GeV monatomic ions.

REVIEWER COMMENTS

Reviewer #3 (Remarks to the Author):

Reviewer #3 (Weixing Li)

The authors have provided significantly improved HR-STEM and EELS images showing the graphite components, rather than their previously considered crystalline diamond, within the C60-induced track in the revised manuscript. Identifying amorphous components is much more challenging than identifying the crystalline diamond because the dominated diamond signals conceal the amorphous ones in the weak core. The authors finally succeeded by choosing a low-index zone axis diamond (100) with a thinner sample thickness (~36 nm) than the ion range (~50 nm) before the ion irradiation and presenting the graphite peak of 285 eV against the distance from the track center in EELS. Besides the possible (but unclear) two-step structure, i.e., a wider hillock or crater followed by a narrower cylindrical track, the authors have also improved the manuscript by mentioning the possibility of the conical ion tracks, i.e., the gradually decreasing track diameter along the incident ion. Therefore, I recommend accepting the publication of this paper in Nature Communications after carefully addressing the minor comments below.

1. P1 Please clarify the intensive π -bonding signal from the graphite in the Abstract.
2. P7 The referenced paper (Dunlop et al., 1997) reported the carrot-shaped track, but the online version did not show the track morphology clearly, possibly due to the photocopying problem. Better cross-sectional TEM images in TiO₂ (Zhai et al., NIMB 457 (2019) 72–79) clearly show the conical ion tracks, i.e., the decreasing track diameter along the incident ion.
3. P8 As the authors mentioned, Fig. 2d is the best side-view image they could obtain. I do not deny that a hillock or crater may exist, but I can only observe a wider track head. The TEM images in Fig. 2 do not show a hillock or a crater on the top of any ion track, as illustrated in Fig.3. The quality of Fig. 2d does not allow one to judge whether a cylindrical track or a conical one follows the wider head. Measurements of track diameter are thus not so reliable. The authors should be more cautious and objective about the related statements despite the improvements of this issue in the revised manuscript.
4. P14 The high-angle annular dark-field STEM image is more reliable than the bright field image in identifying the core/shell structure, as demonstrated in recently reported neutron-induced fission tracks in zircon.
5. P15 The authors did not assign the indices of the diffuse four inner spots in Fig. 4c, although a clear identification of the graphite phase is critical for the conclusion. On the condition that the assignment of diamond spots (022) (Fig. 4c and 4d) is correct, the four inner spots are from the graphite {002} ($d(G002) = 0.3348$ nm). These inner spots cannot be any diamond spots, such as diamond {001} ($d(D001) = 0.3560$ nm), based on the length and angular relation of the R vectors. The authors should add a scale bar in Fig. 4c to help the judgment. Besides the evidence of the diffuse four inner spots in Fig. 4c in the current manuscript, the existence of a graphite phase within the track center is evident because the EELS shows the decreasing graphite signals from the track center against the distance and because Fig. 3d in the last manuscript demonstrates a diffuse ring with two spots belonging to graphite (200). Although one may

distinguish the amorphous graphite from the crystalline one either by the amorphization features within the track in the HR-STEM image or the diffuse ring in the FFT image, estimating the ratio between crystalline and amorphous graphite is difficult. In addition, should the author remove the “and” in the “by filtering and the inverse FFT” in the caption of Fig. 4?

6. P17 The second 295 eV should be 285 eV.

7. P28 Please clarify the exact orientation of the single crystalline diamond [100].

Reviewer #4 (Remarks to the Author):

The results nicely show the formation of ion tracks in diamond which is a first and an important contribution to the field. Compared to the previous version the authors have now employed molecular dynamics simulations to simulate the ion tracks and they appear to be in good agreement with the experimental results. The authors have also used single crystalline diamond to study the track composition and have observed graphitisation. In general I think the extra work has significantly improved the manuscript. I have a couple of comments that I feel need to be addressed.

There is no discussion about the possible synergy of nuclear and electronic stopping. With the cluster irradiation there is a significant component of nuclear stopping which may have a large influence on the track formation process. I understand that the MD simulations do not include nuclear stopping but is that sufficient to rule out a synergy? Synergies have previously been reported (Toulemonde et al. Physical Review B 83 (5), 054106). This will need to be critically discussed. Are tracks only observed because of the velocity effect?

Fig 5 b shows EELS spectra which indicate SP2 bonding. Given the two standards shown as well, can the authors fit the amount of SP2 and SP3 bonding in these spectra?

One thing that I find important is how the track radius/diameter is determined from the images. In particular looking at Fig 4 this seems important as the values can vary significantly depending on how the track boundary is defined in the images. The values are important now as they are compared to MD simulations. I think there should be an example and a description how the track boundary was set in the TEM analysis as well as how many tracks were counted.

For the tracks in the 36 nm thick film the authors note there may be a core shell structure. The samples were thinned down before irradiation. Essentially this means one cannot talk about a bulk track here at these thicknesses. How much of the visible track is part of a hillock or crater (as mentioned before)? In fact, looking at the TEM images in Fig 4, they almost look like craters. Could this explain the low density in the centre?

In Figure 6 a and b (molecular dynamics simulations), it is not clear what is shown, i.e. what do the colors indicate in the images? Displacements? And how is the track boundary defined in the images for

comparison with the experiment. Maybe it would be best to draw the boundary into an example image.

Lastly, now the authors say a 'conventional' thermal spike model does not apply, thus they use MD simulations. The only thing that does not apply (in my opinion) is the melting criterion for defining the track radius. Essentially the MD simulations use the thermal spike model as an input and instead of using the equilibrium melting temperature to define the track boundary the response of the material to the heat spike is simulated (which is a much better approach). I think this needs to be clear in the manuscript so there is no confusion.

One-to-one Response to Reviewers' comments

We express our gratitude to the editor and the reviewers for generous allocation of time and their valuable input in the form of insightful comments and suggestions.

Reviewer #3 :

C3-0) The authors have provided significantly improved HR-STEM and EELS images showing the graphite components, rather than their previously considered crystalline diamond, within the C₆₀-induced track in the revised manuscript. Identifying amorphous components is much more challenging than identifying the crystalline diamond because the dominated diamond signals conceal the amorphous ones in the weak core. The authors finally succeeded by choosing a low-index zone axis diamond (100) with a thinner sample thickness (~36 nm) than the ion range (~ 50 nm) before the ion irradiation and presenting the graphite peak of 285 eV against the distance from the track center in EELS. Besides the possible (but unclear) two-step structure, i.e., a wider hillock or crater followed by a narrower cylindrical track, the authors have also improved the manuscript by mentioning the possibility of the conical ion tracks, i.e., the gradually decreasing track diameter along the incident ion. Therefore, I recommend accepting the publication of this paper in Nature Communications after carefully addressing the minor comments below.

A3-0) Thank you for various important comments and recommending the publication of this paper to Nature Communications.

C3-1) P1 Please clarify the intensive π -bonding signal from the graphite in the Abstract.

A3-1) The words “ π -bonding signal” were replaced by “ π -bonding signal from graphite” in the Abstract.

C3-2) P7 The referenced paper (Dunlop et al., 1997) reported the carrot-shaped track, but the online version did not show the track morphology clearly, possibly due to the photocopying problem. Better cross-sectional TEM images in TiO₂ (Zhai et al., NIMB 457 (2019) 72–79) clearly show the conical ion tracks, i.e., the decreasing track diameter along the incident ion.

A3-2) The authors found a printed copy of Dunlop's paper in their library, in which the TEM images were clearly shown. The sentence was revised as “Another possible candidate was conical ion tracks, which were observed in Y₃Fe₅O₁₂ irradiated with 40.2 MeV C₆₀ ions [27] and in TiO₂ with 1390 MeV Bi ions [28], where the track diameters decreased with the depth.”

C3-3) P8 As the authors mentioned, Fig. 2d is the best side-view image they could obtain. I do not deny that a hillock or crater may exist, but I can only observe a wider track head. The TEM mages in Fig. 2 do not show a hillock or a crater on the top of any ion track, as illustrated in Fig.3. The quality of Fig. 2d does not allow one to judge whether a cylindrical track or a conical one follows the wider head. Measurements of track diameter are thus not so reliable. The authors should be more cautious and objective about the related statements despite the improvements of this issue in the revised manuscript.

A3-3) As shown in Fig. 3R-1, the authors have found that protrusions at the protection-layer side of the Pt surface marker. Judging from the image intensity, the protrusions are made of Pt. However, since the samples were irradiated with C_{60} ions *before* the deposition of Pt on the surface, the protrusions reflect the shapes of the irradiated surface, probably the shapes of the hillocks of the diamond surface. Each Pt protrusion locates on the extension line of each ion track. Therefore, the hillocks of diamond exist at the entrance on the tracks. The Fig. 3R-1 was added to Fig. 2 panel in text.

Of course, this is indirect evidence since the authors observed protrusions made of Pt, not directly the hillocks. The authors try to be more cautious and objective about the related statements, as suggested by the reviewer.

Fig. 3R-1. A BF-TEM side-view image of diamond irradiated with 9 MeV C_{60}^{2+} ions. Thick black layer is a deposited Pt film as surface marker. Many protrusions are observed at the protection-layer side of the surface maker. While the observed protrusions are made of Pt, the origin of the protrusions can be the hillocks made by the ion tracks.

Sentences in the section “Shape and dimensions of tracks, and S_e threshold” were revised:

“Figures 2c-e show side-views of the tracks in low, medium, and high magnifications, respectively. Thick black layers are deposited Pt films for the surface markers, and the

tracks are observed at one side of the markers. As shown in Fig. 2d, many protrusions observed at the protection-layer side of the Pt marker. Judging from the image intensity, the protrusions are made of Pt. However, since the Pt layers were deposited *after* the irradiations to diamond with C_{60} ions, the protrusions reflect the shapes of the irradiated surface, probably the shapes of the hillocks on the diamond surface. This assumption is supported by the observations that each Pt protrusion locates on the extension line of each ion track. However, it should be noted that this is indirect evidence of the diamond hillocks since we observed Pt protrusions only.

It should be noted that higher density of the tracks is observed in Fig. 2c than in Fig. 2d, probably because the sample of Fig. 2c is thicker. Because of the higher density of the tracks, higher density of the Pt protrusions is expected. Because of the overlap due to the high density, the protrusions are not clearly distinguished with each other in Fig. 2c.

Figure 2e exhibits an expanded image of two ion tracks. Since this figure is not clear, one cannot judge whether cylindrical or conical tracks. Therefore, the observed tracks with narrower tails (Fig. 2b) could be ascribed to surface-craters [29], hillocks [30] or conical tracks [27,28]. As shown later, a track with three legs was observed by HR-STEM. To explain the track shape, we assume the material emission from the track and deposition on the sample surface, which could be consistent with the hillocks and craters.”

C3-4) P14 The high-angle annular dark-field (HAADF STEM) image is more reliable than the bright field image in identifying the core/shell structure, as demonstrated in recently reported neutron-induced fission tracks in zircon.

A3-4) While the authors called the core/shell tracks in the previous manuscript, now they assume that the tracks were formed with material emission from the track core and the deposition of the material to the peripheries of the tracks. A HAADF image is shown in Fig. 3R-2. While the high-resolution images in the previous manuscript were observed

Fig. 3R-2. A HAADF image of the ion track formed in diamond irradiated with 9 MeV C_{60} ions, detected by a low-resolution STEM.

by an aberration corrected HR-STEM, the HR-STEM belongs to common-used facilities and is fully occupied by other reservations. The authors need to wait for a couple of months. Therefore, HAADF observation was conducted using an old low-resolution STEM with a spot size of 0.5 nm. Figure 3R-2 shows a black track core indicating a lower density or lesser thickness, which is surrounded by a slightly white ring indicating a higher density of greater thickness. This figure was added in the supplementary material S10.

C3-5) P15 The authors did not assign the indices of the diffuse four inner spots in Fig. 4c, although a clear identification of the graphite phase is critical for the conclusion. On the condition that the assignment of diamond spots (022) (Fig. 4c and 4d) is correct, the four inner spots are from the graphite {002} ($d(G002) = 0.3348$ nm). These inner spots cannot be any diamond spots, such as diamond {001} ($d(D001) = 0.3560$ nm), based on the length and angular relation of the R vectors. The authors should add a scale bar in Fig. 4c to help the judgment. Besides the evidence of the diffuse four inner spots in Fig. 4c in the current manuscript, the existence of a graphite phase within the track center is evident because the EELS shows the decreasing graphite signals from the track center against the distance and because Fig. 3d in the last manuscript demonstrates a diffuse ring with two spots belonging to graphite (200). Although one may distinguish the amorphous graphite from the crystalline one either by the amorphization features within the track in the HR-STEM image or the diffuse ring in the FFT image, estimating the ratio between crystalline and amorphous graphite is difficult. In addition, should the author remove the “and” in the “by filtering and the inverse FFT” in the caption of Fig. 4?

A3-5a) The authors have added a scale bar in Fig. 4c, and have confirmed the diffuse four inner spots in Fig. 4c can be ascribed to graphite {002}. The following sentences were added to describe the formation of crystalline graphite phase from the FFT pattern. Co-existence of the amorphous cores and the crystalline graphite was also described.

“While Fig. 4c shows bright spots of 022- and 004-related diffractions from the diamond lattice, four diffuse spots are observed much closer to the center spot than the diamond D022 spots. These spots cannot be ascribed to the diamond but to the graphite G002 and G00-2. Since G002 and G00-2 appear as a pair of spots, the observation of the four spots indicates the existence of crystalline graphite grains of different orientations. HR-STEM observations show both the formation of amorphous regions at the track cores and of crystalline phase of graphite.”

Also, the caption of Fig. 4 was corrected as suggested by the reviewer. Thank you.

C3-6) P17 The second 295 eV should be 285 eV.

A3-6) It was a typo and corrected. Thank you.

C3-7) P28 Please clarify the exact orientation of the single crystalline diamond [100].

A3-7) The description of the sample preparation in the method section was revised to clarify the orientation of the single crystalline diamond [100].

“Sample preparation. Poly-crystalline and single-crystalline diamond self-standing samples, both of which were grown by the chemical vapor deposition (CVD) technique, were purchased from Element Six Co. At first, the poly-crystalline samples were utilized for all the experiments, then the single-crystalline samples of 3 mm × 3 mm × 0.25 mm in dimensions were utilized for further HR-STEM and STEM-EELS measurements. The crystal face of 3 mm × 3 mm corresponded to the [1 0 0] of diamond, while the other faces of 3 mm × 0.25 mm corresponded to the [0 1 0] and [0 0 1].”

Reviewer #4 (Remarks to the Author):

C4-0) The results nicely show the formation of ion tracks in diamond which is a first and an important contribution to the field. Compared to the previous version the authors have now employed molecular dynamics simulations to simulate the ion tracks and they appear to be in good agreement with the experimental results. The authors have also used single crystalline diamond to study the track composition and have observed graphitisation. In general I think the extra work has significantly improved the manuscript. I have a couple of comments that I feel need to be addressed.

A4-0) Thank you for careful reading.

C4-1a) There is no discussion about the possible synergy of nuclear and electronic stopping. With the cluster irradiation there is a significant component of nuclear stopping which may have a large influence on the track formation process. I understand that the MD simulations do not include nuclear stopping but is that sufficient to rule out a synergy? Synergies have previously been reported (Toulemonde et al. Physical Review B 83 (5), 054106). This will need to be critically discussed. Are tracks only observed

because of the velocity effect?

A4-1) According to the present MD simulations, the experimental results are roughly explained by the velocity effect only, without assuming the synergy effect additionally. However, it should be noted that the authors have carried out a few of approximations or simplifications to apply the TT-MD simulations to a cluster ion impact.

(a) In the case of C_{60} ion impact, each C atom consisting of a C_{60} molecule is injected to different position within the size of C_{60} molecule (the radius of 0.35 nm). Since the 66%-energy-deposition radius for each C ion was 0.4 nm, the cooperative energy deposition is possible between C atoms at different sites. However, this effect probably results in the enlargement of the excitation volumes, i.e., the reduction of the excitation density. Therefore, the authors expected underestimation of the track formation in diamond by the MD simulations. However, since the experimental results were well-reproduced by the MD simulations, the authors are looking for another factor enhanced the excitation density to compensate the effect of the multiple injection points.

While it is a purely speculation, the synergy effect between S_e and S_n cannot be excluded. Following sentences were added in the Discussion.

“However, it should be noted that an approximation was provided to apply the TT-MD simulations to a cluster ion impact. While each C atom consisting of a C_{60} molecule is injected to different position within the size of the C_{60} molecule (the radius of 0.35 nm), sixty C atoms are approximated to be injected to the same position in the simulations. Since the radius in which 66% of the electronic energy is stored is 0.4 nm for each C ion in the present case, the cooperative energy deposition is possible between C atoms at different sites. However, this effect probably results in the enlargement of the excitation volumes, i.e., the reduction of the excitation density. The present simulations should overestimate the track radii. However, the experimental results were well-reproduce by the simulations. We cannot exclude other effects such the synergy effect between S_e and S_n to enhance the track radii.”

C4-2) Fig 5 b shows EELS spectra which indicate SP2 bonding. Given the two standards shown as well, can the authors fit the amount of SP2 and SP3 bonding in these spectra?

A4-2) Thank you. It is interesting. The result is shown in Fig. 3R-3.

A blue curve in Fig. 3R-3 shows the EELS spectrum detected at the center of the track. The experimental curve was fitted by a sum of the reference data of graphite and diamond with changing the concentration ratio y of the two materials,

$$I(E) = y I_G(E) + (1 - y) I_D(E).$$

The best fitting was obtained with the ratio $y = 30\%$, which indicated that non-negligible content of graphite or at least π -bondings exist in the track cores. This observation could be consistent with the observation of broad graphite peak in an FFT-image shown in Fig. 4c.

However, the spot size of our EELS was 0.5 nm, and the tracks are short enough to reach the other side of samples. We have measured the EELS spectrum shown in Fig. 3R-3 at the center of the track. Even at the center of the track, the diamond phase of only 30% transforms to the graphite phase. The 70% are not transformed to graphite after a C_{60} ion impact.

Fig. 3R-3. (upper) The blue curve indicates the STEM-EELS spectrum of diamond irradiated with 9 MeV C_{60} ions, measured at the center of the ion track. The experimental spectrum was fitted with the sum of the reference spectra of graphite and diamond with the optimal ratio of 0.3 as shown by a red curve.

Figure 3R-3 was added to the panel of Fig. 5 in text, and following sentences were added: “The observed EELS spectra $I(E)$ were fitted as the sum of the standard data of graphite and diamond, $I_G(E)$ and $I_D(E)$, which are shown at the bottom of Fig. 5b, by optimizing with the graphite ratio y : i.e.

$$I(E) = y I_G(E) + (1 - y) I_D(E),$$

where $0 < y < 1$.

From the fitting, the graphite ratio y was determined. The ratio was the highest at the center of the track, which reached $y = 0.3$ at the center. The ratio y decreased with the distance from the center of the track.

This indicates that non-negligible content of graphite or at least π -bondings exist in the track cores. This observation could be consistent with the observation of broad graphite peak in an FFT-image shown in Fig. 4c. However, the spot size of our EELS was 0.5 nm, and the tracks are connected to the other side of the samples. We have measured the EELS spectrum shown in Fig. 5d at the center of the track. Even at the center of the track, the diamond phase of only 30% transforms to the graphite phase. The 70% are not transformed to graphite after a C_{60} ion impact.”

C4-3) One thing that I find important is how the track radius/diameter is determined from the images. In particular looking at Fig 4 this seems important as the values can vary significantly depending on how the track boundary is defined in the images. The values are important now as they are compared to MD simulations. I think there should be an example and a description how the track boundary was set in the TEM analysis as well as how many tracks were counted.

A4-3) While the authors called core-shell tracks from the high-resolution top view in the previous manuscript, now we consider that the strange shapes of the tracks are due to material emission from the track and inhomogeneous deposition. The core of a track becomes lower dense or less thick due to the material emission during the track formation. The emitted material deposited at the peripheries of the track, resulting in slightly thicker inhomogeneous regions.

See Fig. 3R-4. This figure shows protrusions on the surface marker (Pt). Judging from the image intensity, the protrusions are made of Pt. However, these structures probably reflect the shapes of the irradiated diamond surface, i.e., the hillocks of diamond.

As already shown in Fig. 3(b), the mean track diameter observed by the top-view is always larger than that observed by the side-view.

Therefore, the authors used the mean track diameters determined from the side-view, rather than those from the top-view. Since the high-resolution top-views show some strange shapes, it is not easy to distinguish the true track part from the deposited material.

Fig. 3R-4. A BF-TEM side-view image of diamond irradiated with 9 MeV C_{60}^{2+} ions. Thick black layer is a deposited Pt film as surface marker. Many protrusions are observed at the protection-layer side of the surface marker. While the observed protrusions are made of Pt, the origin of the protrusions can be the hillocks made by the ion tracks.

C4-4a) For the tracks in the 36 nm thick film the authors note there may be a core shell structure. The samples were thinned down before irradiation. Essentially this means one cannot talk about a bulk track here at these thicknesses.

A4-4a) The authors have changed the model to explain the strange shapes of the tracks observed by HR-STEM, from the core-shell model to the material emission model and deposition. In the 36 nm-thick sample, the tracks reach the other side of the sample. This could be a very different situation from the tracks formed in bulk samples. However, this thin sample thickness is important to judge whether amorphous regions form in the track or not. Following sentence was added in the section “HR-STEM observation.

“This quite thin thickness is an important condition to judge whether amorphous regions form in the tracks or not, while the track formation in such thin layer could be slightly different from that in bulk.”

C4-4b) How much of the visible track is part of a hillock or crater (as mentioned before)?

A4-4b) Judging from the tilting images similar to Fig. 2b, majority (more than ~50%) of tracks showed the two-step structures. From Fig. 3R-4, the mean height of the hillocks is less than 10 nm, while the mean track length is ~50 nm.

C4-4c) In fact, looking at the TEM images in Fig 4, they almost look like craters. Could this explain the low density in the centre?

A4-4c) The track center in lower density or less thickness was formed due to the material emission from the track. The emitted material deposits at the peripheries of the track, resulting in inhomogeneous thicker regions. The emitted material was observed as

hillocks.

C4-5a) In Figure 6 a and b (molecular dynamics simulations), it is not clear what is shown, i.e. what do the colors indicate in the images? Displacements?

A4-5a) Unfortunately, image compression had lowered the quality of the images, making them harder to interpret. There is no color coding in the figure. These are atoms drawn with the ball-and-stick model using a single color (blue). Shades appear since the figures were generated using photorealistic rendering program. In a high-resolution image, these shades guide the eye. Defects and amorphous or strained regions appear darker. As an example, shown below is a high-resolution version of the fullerene track at 60 keV/nm.

Fig. 3R-5. A high-resolution image of the track in diamond. Carbon atoms and bondings are shown by the single ball-and-stick model using a single color of blue. The shades are due to a photorealistic rendering function of the plotting program.

However, figure 6 is also too small to display these details. To make it easier to understand what is depicted in the figure, we have redrawn the images without the photorealistic renderer. Atoms are now represented simply as dots, bonds are not drawn, the aggressive image compression is turned off, and the font size is largened. We have also added an explanation to the caption of figure 6 that helps understanding the figure.

C4-5b) And how is the track boundary defined in the images for comparison with the experiment. Maybe it would be best to draw the boundary into an example image.

A4-5) As shown in Fig. 6 in the text, all the simulated tracks are nearly spherical, which does not reproduce the strange shape observed by STEM observation. While at the moment the authors assume that the strange shape is due to the material emission and re-deposition, these phenomena are not expected in the present code. A sentence was added: “The simulated tracks are almost spherical, which is nearly consistent with low-magnification BF-TEM images (Figs. 1b and 2a) but not with high-magnification STEM image (Fig. 4).”

The reported values for the MD simulations were derived from visualizations shown in Fig. 6 by eye. The cylindrical ion tracks are seen from the visualizations as dark areas in the middle of the top view. This is because in an undistorted lattice, atoms form columns along the visualization direction, creating white space between them. In an amorphous or severely defected zone however, there is no similar columnar structure, resulting in a darker appearance.

We have written a simple computer software that can be used to place markers on the figures. Markers were placed on opposite sides of the cylindrical regions by eye, which can be used to estimate the radii. The errors were determined using the sample variance of the radii. While this method is inherently approximate, it aligns with the intended purposes of the paper. We are happy to provide the simulation cells and the software if necessary, or an alternative way to determine the track radii (using e.g., density fluctuations).

We have clarified the description of track radii measurement in section Methods, Two-temperature molecular dynamics simulations.

C4-6) Lastly, now the authors say a 'conventional' thermal spike model does not apply, thus they use MD simulations. The only thing that does not apply (in my opinion) is the melting criterion for defining the track radius. Essentially the MD simulations use the thermal spike model as an input and instead of using the equilibrium melting temperature to define the track boundary the response of the material to the heat spike is simulated (which is a much better approach). I think this needs to be clear in the manuscript so there is no confusion.

A4-6) The following sentence was added in the end of Introduction:

“One of the Advantages of the TT-MD simulations is no requirement of the input of the equilibrium melting temperature.”

Also, the following sentences in Discussion were revised:

” Here we have applied the two temperature molecular dynamics (TT-MD) simulations: This model describes the motions of C atoms interacting via the interatomic potentials under rapid energy deposition from the electronic system, without assuming any phase transitions such as the melting of diamond or graphitization. This is in contrast with the i-TS model, where the phase transition temperature, mostly the melting temperature at the equilibrium, should be input. Furthermore, the TT-MD simulations could detect the track formation without phase transitions, i.e., without melting or vaporization.”

REVIEWER COMMENTS

Reviewer #3 (Remarks to the Author):

The authors have addressed all the concerns in my last review and modified the manuscript accordingly. I recommend the acceptance of this paper in Nature Communications.

In the revised manuscript, the authors used many protrusions at the protection-layer side of the surface maker (~20 nm thick) to support the hillocks (less than 5 nm high) on the diamond surface. This explanation is not easy to understand without a better example. The authors can cite their paper (Amekura et al., Quantum Beam Sci. 2022, 6, 4) to help the readers judge the relationship between the protrusions and hillocks from Fig. 2b of the cited paper. Please add the missing information (thickness or composition) on the Pt marker layer and the additional protection layer.

Reviewer #4 (Remarks to the Author):

I would like thank the authors for addressing most of my comments and the comments of the other referee. Two of my comments have not been addressed satisfactorily and I would still recommend to address these before publication:

I asked how the track diameters were determined, i.e. how the boundaries were set in the TEM images to get the results. This has not been explained in the revision. This is important however, for example in Fig 2 it appears that there is variation along the track and different tracks have different widths. Setting the boundaries is somewhat arbitrary and it is important to know where and how the boundaries for the measurements have been set. This should be explained in detail, best with showing an example, in the supplementary information. The authors mention that the track diameters were determined from the side view images but only side view images for 9 MeV irradiation are shown. It is essential to include side view images from all the other energies as well (can be in supplementary information) so the reader can verify the results and track radii. This is central to the paper so can't be omitted.

Secondly I had suggested fitting the EELS spectra fit to the standards which has been done for one spectrum. Why not do all in Fig 5? Then one could assess if that is a viable method yielding consistent results.

One-to-one Response to Reviewers' comments

We greatly appreciate the editor and the reviewers for kind comments and suggestions.

Reviewer #3 (Remarks to the Author):

The authors have addressed all the concerns in my last review and modified the manuscript accordingly. I recommend the acceptance of this paper in Nature Communications.

C3-1) In the revised manuscript, the authors used many protrusions at the protection-layer side of the surface marker (~20 nm thick) to support the hillocks (less than 5 nm high) on the diamond surface. This explanation is not easy to understand without a better example. The authors can cite their paper (Amekura et al., Quantum Beam Sci. 2022, 6, 4) to help the readers judge the relationship between the protrusions and hillocks from Fig. 2b of the cited paper. Please add the missing information (thickness or composition) on the Pt marker layer and the additional protection layer.

A3-1) Thank you for suggestions. The thickness of Pt layer and the composition of the protection layer were specified. The paper specified by the Reviewer was referred. Following sentences were added.

“Thick black layers are deposited Pt films for the surface markers of ~20 nm in thickness, and the tracks are observed at one side of the Pt markers. To protect the Pt marker against the focused ion beam (FIB) milling for sample thinning, thick carbon layer was deposited over it.”

“While the Pt thickness was thicker than the heights of the hillocks, the hillocks of quartz crystals have been successfully detected by this method [24].”

Reviewer #4 (Remarks to the Author):

I would like thank the authors for addressing most of my comments and the comments of the other referee. Two of my comments have not been addressed satisfactorily and I would still recommend to address these before publication:

C4-1) I asked how the track diameters were determined, i.e. how the boundaries were set in the TEM images to get the results. This has not been explained in the revision. This is important however, for example in Fig 2 it appears that there is variation along the track and different tracks have different widths. Setting the boundaries is somewhat arbitrary and it is important to know where and how the boundaries for the measurements have been set. This should be explained in detail, best with showing an example, in the supplementary information. The authors mention that the track diameters were determined from the side view images but only side view images for 9 MeV irradiation are shown. It is essential to include side view images from all the other energies as well (can be in supplementary information) so the reader can verify the results and track radii. This is central to the paper so can't be omitted.

A4-1) We used a very primitive method to set the boundary of a track. See Fig. 4R1. At first, the contrast of the image was optimized. (We confirmed this procedure does not the track diameter largely.) After then, a track was sandwiched by two parallel lines as shown in Fig. 4R1, and the diameter was determined. In the case of the tracks whose diameter changes with the depth, a kind of averaging via human's eye was done, and "averaged diameter along the depth" was determined. A following sentence was added in the text near Fig. 3 to invite the Supplementary Information S11. "Regarding the determination of the mean track diameter from the side-view images, see Supplementary Information S11."

In the Supplementary Information S11, side-view images of diamond samples irradiated with 2, 4, 6, and 9 MeV C₆₀ ions are shown.

C4-2) Secondly I had suggested fitting the EELS spectra fit to the standards which has been done for one spectrum. Why not do all in Fig 5? Then one could assess if that is a viable method yielding consistent results.

A4-2) We have fitted all the spectra in Fig. 5. The graphite ratio y was 0.30 at the center of the track, and decreased with increasing the distance from the center. This distance dependence of the graphite ratio was plotted in Fig. 5(c).

Fig. 4R1. A side view of the bright-field TEM image of diamond sample irradiated with 9 MeV C₆₀ ions. The image contrast was modified. A pair of lines near the right edge indicates how to determine the track diameter.

REVIEWERS' COMMENTS

Reviewer #3 (Remarks to the Author):

The authors have addressed my comments on the Pt marker layer and modified the manuscript accordingly. I have no further comments.

Reviewer #4 (Remarks to the Author):

The authors have addressed the remaining comments. Given the new information I am puzzled how the authors can determine a track diameter from Fig. S12a (2 MeV) with an uncertainty of $\sim \pm 0.5$ nm (Fig 3 b). The tracks are very faint and clearly not cylindrical. However, as the track diameter is not critically important for the conclusions I don't suggest any further revision.

Answer to Optional Reviewer comment

While all the reviewers have agreed the publication of this manuscript, we have further revised the manuscript following the last comment shown below.

Reviewer #3 (Remarks to the Author):

The authors have addressed my comments on the Pt marker layer and modified the manuscript accordingly. I have no further comments.

Authors: Thank you.

Reviewer #4 (Remarks to the Author):

The authors have addressed the remaining comments. Given the new information I am puzzled how the authors can determine a track diameter from Fig. S12a (2 MeV) with an uncertainty of $\sim +0.5$ nm (Fig 3 b). The tracks are very faint and clearly not cylindrical.

However, as the track diameter is not critically important for the conclusions.

I don't suggest any further revision.

Authors: Following sentences were added in the Supplementary Note 10 “Particular in the samples irradiated with 2 MeV ions, some tracks look not perfectly cylindrical. However, we have determined the diameters assuming the perfect cylinders, and then the determined mean diameter and SD were shown in Fig. 3c in the main text. The error bars were simply due to the SD, i.e., the statistical scattering of the diameters. The uncertainty due to the non-perfect cylinder shapes is not taken into account.”